# Dormant origin firing promotes head-on transcription-replication conflicts at transcription termination sites in response to BRCA2 deficiency

Liana Goehring[1], Sarah Keegan[1,2], Sudipta Lahiri[1,3], Wenxin Xia[1], Michael Kong[1], Judit Jimenez-Sainz [3], Dipika Gupta [1], Ronny Drapkin [4,5], Ryan B. Jensen [3], Duncan J. Smith [6], Eli Rothenberg [1], David Fenyö[1,2] & Tony T. Huang [1] ✉

BRCA2 is a tumor suppressor protein responsible for safeguarding the cellular genome from replication stress and genotoxicity, but the specific mechanism(s) by which this is achieved to prevent early oncogenesis remains unclear. Here, we provide evidence that BRCA2 acts as a critical suppressor of head-on transcription-replication conflicts (HO-TRCs). Using Okazaki-fragment sequencing (Ok-seq) and computational analysis, we identified origins (dormant origins) that are activated near the transcription termination sites (TTS) of highly expressed, long genes in response to replication stress. Dormant origins are a source for HO-TRCs, and drug treatments that inhibit dormant origin firing led to a reduction in HO-TRCs, R-loop formation, and DNA damage. Using super-resolution microscopy, we showed that HO-TRC events track with elongating RNA polymerase II, but not with transcription initiation. Importantly, RNase H2 is recruited to sites of HO-TRCs in a BRCA2-dependent manner to help alleviate toxic R-loops associated with HO-TRCs. Collectively, our results provide a mechanistic basis for how BRCA2 shields against genomic instability by preventing HO-TRCs through both direct and indirect means occurring at predetermined genomic sites based on the pre-cancer transcriptome.

High-grade serous ovarian carcinoma (HGSOC) is the most common type of ovarian cancer and the deadliest gynecologic malignancy[1]. Recent genomic studies have revealed that the distal fallopian tube epithelium (FTE) is the likely cell-of-origin for HGSOC[2–5]. These ovarian cancer precursor cells have undergone malignant transformation driven primarily by mutations in *TP53*, and are characterized by high DNA damage load, evidenced by high gH2AX staining in fallopian tube fimbrae[3,5]. Despite progress towards understanding the etiology of ovarian cancers, the endogenous sources of DNA damage that drive genomic instability and cancer progression in FTEs remain unclear. Nearly half of HGSOC are homologous recombination (HR)-deficient, including germline or somatic loss-of-function mutations in the *BRCA2* gene[6]. In addition to its canonical role in HR DNA repair, BRCA2 also mitigates genotoxic lesions caused by increased replication stress through the protection of replication-fork integrity[7–12].

[1]Department of Biochemistry & Molecular Pharmacology, New York University School of Medicine, New York, NY, USA. [2]Institute for Systems Genetics, New York University School of Medicine, New York University School of Medicine, New York, NY, USA. [3]Department of Therapeutic Radiology, Yale University, New Haven, CT, USA. [4]Penn Ovarian Cancer Research Center, University of Pennsylvania, Perelman School of Medicine, Philadelphia, PA, USA. [5]Basser Center for BRCA, Abramson Cancer Center, University of Pennsylvania, Perelman School of Medicine, Philadelphia, PA, USA. [6]Center for Genomics and Systems Biology, Department of Biology, New York University, New York, NY, USA. ✉e-mail: tony.huang@nyumc.org

Transcriptional-associated replication stress and DNA damage are caused, in part, by persistent R-loops; R-loops are RNA:DNA hybrids that form spontaneously between the RNA and DNA template during transcription, DNA repair, and transcription-replication collisions[13–20]. Recent work has shown that R-loop resolution and mitigation of R-loop-associated DNA damage is facilitated by several RNA-binding factors, including RNase H2, an RNA-DNA hybrid-specific nuclease that binds RNA polymerase II (RNAP2) and prevents R-loop accumulation during transcriptional elongation[21]. Recent reports have implicated BRCA2 in the regulation of transcription and in the recruitment of RNase H2 to sites of DNA breaks[22–24]. However, the genomic location and specific mechanisms by which BRCA2 prevents transcription-associated DNA damage and genomic instability in cancers remain unclear.

Effective replication origin firing and replication-fork progression are critical to proper genome duplication, cell cycle progression, and prevention of DNA damage. Tens of thousands of replication origins are licensed in excess throughout the entirety of the genome, but only a subset of these fire to activate forks at euchromatic, transcriptionally active regions in early S-phase[25,26], and passive replication inhibits nearby licensed origins from firing[27]. Replication stress is characterized by defective fork progression due to obstruction, depletion of replication factors, or DNA damage, resulting in excessive single-stranded DNA coated by RPA, and followed by ATR activation[28]. A proper replication stress response to slowed/stalled forks includes the activation of adjacent dormant origins to help compensate for these stressed forks and to prevent under-replication of the genome. Persistent replication stress and failure to activate dormant origins can lead to fork collapse/DNA breaks, genomic instability, incomplete replication, and cancer progression[29–32].

The mechanisms by which replication stress contributes to genomic instability in early BRCA2-mutated ovarian cancers are not well understood. One major source of replication stress linked to transcriptional dysregulation is transcription-replication conflicts (TRCs). Studies in diverse prokaryotic and eukaryotic organisms reveal that TRCs cause replication stress, DNA damage, and mutagenesis[16,33–35]. TRCs can be co-directional (CD-TRC), where the replication fork moves in the same direction as the transcription machinery on the same DNA template to help minimize replication stress, or head-on (HO-TRC), where the two types of machinery move in the opposite direction and converge, leading to the formation of toxic R-loops[36–38]. While both types have been shown to contribute to some level of genomic instability in different biological systems[33,39–44], HO-TRCs have been reported to be associated with persistent R-loops and ATR activation compared to CD-TRCs in a cell model[36]. Indeed, a body of work that uses high-throughput sequencing (HTS) techniques to interrogate sites of active DNA synthesis at the genome-scale reveals that inherent genome organization promotes co-directional movement of replication and transcription machineries[25,27,45–49]. For example, methods that map replication-fork directionality at the genome-wide scale in human cells have demonstrated that replication origins are localized preferentially at transcription start sites (TSSs) of long and highly transcribed genes and that replication termination occurs at these genes' transcription termination sites (TTSs)[27,46,49]. How replication-fork dynamics in the context of BRCA2 deficiency influences the location and enrichment of TRCs, and whether locus-specific TRCs impact genomic instability and cancer development remains ill-defined.

Here, we define a critical role for BRCA2 in genomic maintenance in the context of TRCs and establish the consequences of BRCA2 loss by integrating a multi-pronged experimental approach with computational tools. To identify potential genomic sites of TRCs, we employed Okazaki-fragment sequencing (Ok-seq) that provides the location of replication origins at the genome-wide scale, revealing that newly fired replication origins (dormant origins) preferentially occur near the TTSs of active genes in response to global replication stress. Using an FTE cell line to model BRCA2 deficiency in ovarian cancer precursor cells, we show that the dormant origin firing occurring near TTSs leads to the accumulation of TTS-oriented HO-TRC events. To define the molecular complexes associated with these lesions, we utilized both proximity-ligation assay (PLA) and single-molecule localization microscopy (SMLM-STORM) techniques, demonstrating that defects in fork elongation led to downstream dormant origin firing and enhanced HO-TRCs in BRCA2-deficient cells, which can be partially alleviated using PARP or CDC7 inhibitors. Importantly, BRCA2-dependent recruitment of RNase H2 to sites of HO-TRCs suppresses R-loop formation and transcription-associated DNA damage. Our data provide a mechanistic basis for how BRCA2 shields against genomic instability by preventing HO-TRCs through both direct and indirect means occurring at predetermined genomic sites based on the pre-cancer transcriptome.

## Results

### BRCA2 deficiency triggers new/dormant origin firing at transcription termination sites

Reduced expression and loss of BRCA2 activity lead to neoplasm and transformation due, in part, to the accumulation of replication-associated deleterious events whose origins are not well-defined. To explore this we utilized a relevant cellular system that models the loss of BRCA2 in cells-of-origin that are linked to transformation in HGSOC. We implemented Ok-seq analysis to study how transient BRCA2 deficiency in a cellular model can influence replication origin usage and fork directionality. BRCA2 knockdown (KD) was achieved in immortalized primary human fallopian tube epithelial cells (FTEs or fallopian tube secretory epithelial cells)[50] that were either transduced with TET-on shBRCA2 or shScramble constructs to generate doxycycline (DOX)-inducible KD of BRCA2 or control scrambled KD (Supplementary Fig. 1a). Published work from our group and others have demonstrated the application of Ok-seq analysis for high-resolution, genome-wide interrogation of replication-fork dynamics across the human genome[27,51]. Briefly, Okazaki fragments from untransformed, asynchronous, and actively replicating cells are isolated, captured, and sequenced, as previously described[51]. Following read processing and alignment to the human genome, replication-fork directionality analysis can determine replication origin location and efficiency. We used a new computational methodology to analyze our previous Ok-seq dataset to probe for replication origins from Ok-seq reads mapped to the Watson and Crick strands. Following hanning smoothing of 1 kb-binned Watson and Crick reads, we calculated the discrete derivative (DER) of smoothed reads on both strands, and continuous regions of increasing Crick and decreasing Watson were identified as sites of replication initiation (Fig. 1a, i). The origin efficiency was quantified by the maximum DER-score over the region (see "Methods" section). After normalization, the resulting distribution of scores was negatively skewed, and origins were filtered by applying a fwhm cutoff (Supplementary Fig. 1b). This analysis pipeline generated over 10,000 replication origins per sample (Supplementary Fig. 1c). To determine how origins are oriented across various gene annotations, we used relative distance calculation[52] (Fig. 1a, ii) that allows for genome-wide investigation of spatial correlation and orientation between datasets of interest without strict overlap and consideration of dataset interval frequency. Work from our group and others has shown that efficient origins occur preferentially at TSSs in a transcription-dependent manner[25,27,46]. Indeed, origin score and relative distance analysis to TSSs show a strong bias of origins that overlap with transcriptionally active TSSs, many that are near active TSSs, and some that are not oriented around TSSs (Fig. 1a). If there was a

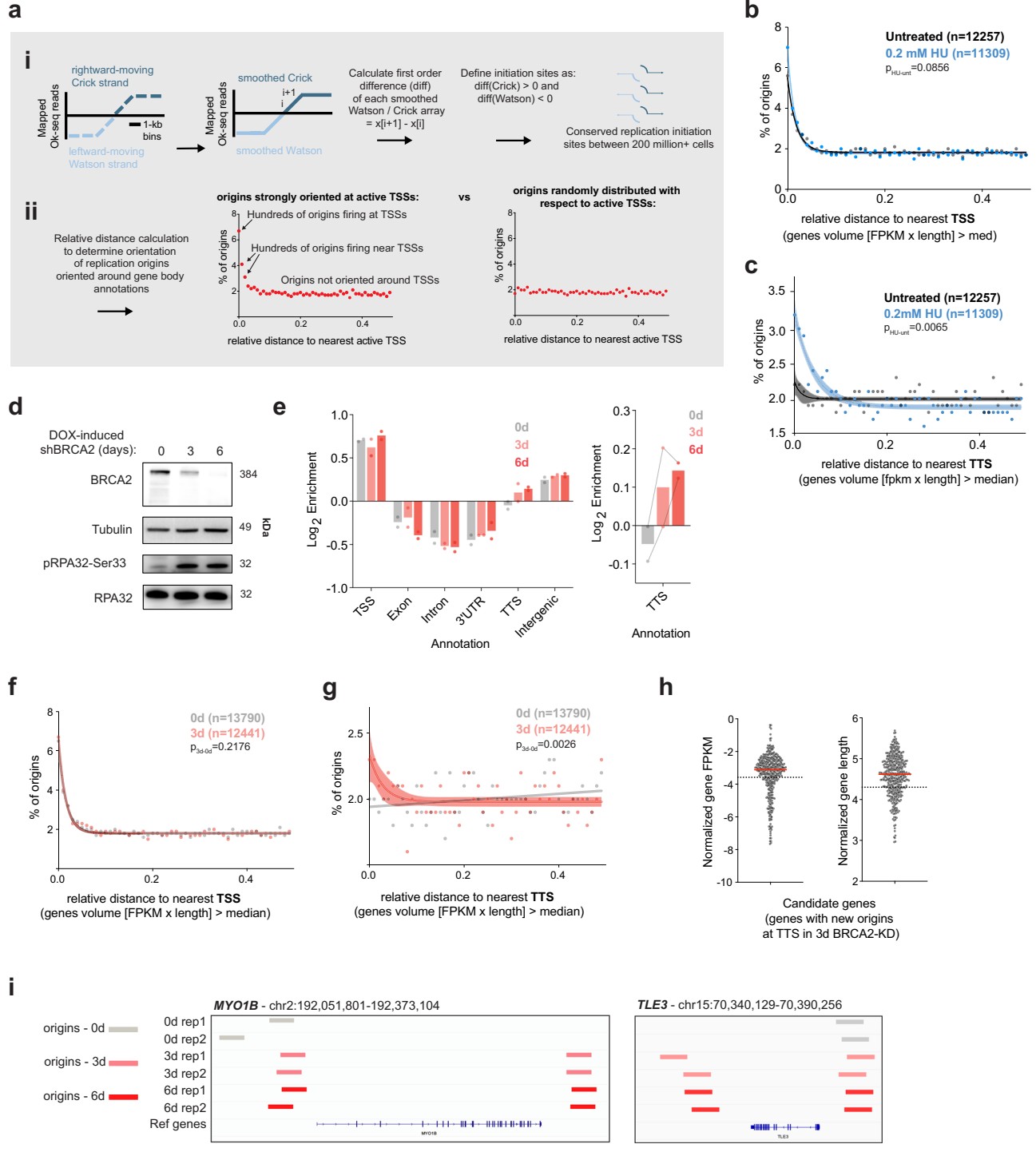

random distribution of origins with respect to TSSs, we expect proportion of origins with a relative distance at or nearby active TSSs to be equal to proportion of those not oriented near an active TSS (Fig. 1a, ii). To validate this analysis, we used our previously published Ok-seq results in untreated and HU-treated RPE-1 cells[27]. Moderate nucleotide depletion in the setting of hydroxyurea (HU) treatment slows replication forks and causes compensatory dormant origin firing[53,54]. In our previous study, mapping replication-fork directionality in HU-treated cells revealed a global increase in origin firing at both canonical TSSs and downstream of TTSs of transcriptionally active genes[27]. We previously identified "transcriptional volume", defined as the *number of RNAs (FPKM) multiplied by gene length*, and therefore indicative of the number of RNA

polymerase molecules occupying a gene, as a strong correlate of TSS-proximal replication initiation. Relative distance calculations across gene bodies showed that HU-treated RPE-1 cells have increased origin firing at TSSs of high-volume genes (Fig. 1b), and a significant increase in origins oriented at TTSs of high-volume genes compared to untreated conditions (Fig. 1c); a representative locus of untreated and HU-treated RPE-1 Ok-seq datasets are shown (Supplementary Fig. 1d).

To determine the best time-point to evaluate the replication defects caused by BRCA2 deficiency, we measured BRCA2-KD (knockdown-down) levels after three and six days of DOX treatment (Fig. 1d). Consistent with previously described roles for BRCA2 in preventing replication stress[55], loss of BRCA2 showed increase

**Fig. 1 | Loss of BRCA2 causes dormant origin firing at TTSs of long, highly transcribed genes. a** Identification of new origins at TTSs upon replication stress and schematic of Okazaki-fragment sequencing origin analysis. (i) 1 kb-binned Crick and Watson Ok-seq reads are Hann smoothed and replication origins calls are defined as regions of increasing Crick-strand derivative and decreasing Watson-strand derivative from smoothed reads (see "Methods" section). (ii) Relative distance calculation orients origins to genomic annotations (TSS, TTS). Replication origins are prevalent at active TSSs indicated by a high proportion of origins at low relative distances in active TSSs (left). If origins were randomly distributed with respect to active TSSs, there would be an equal proportion at all relative distances (right). **b** Relative distance origin analysis of previously published untreated and HU-treated RPE-1 cells (Chen et al.[27]) relative to TSSs of transcriptionally high-volume genes (FPKM × length > median). $N$ = # of origins per condition. Percent of origins at each relative distance were fit to a nonlinear regression curve (solid line) with a 90% confidence interval (shaded). $p$-values calculated using a one-sided, paired $t$-test between conditions for the first three relative distance bins (0, 0.01, 0.02) (see "Methods" section). **c** Relative distance origin analysis of untreated and HU-treated RPE-1 cells relative to TTSs of high-volume genes. The solid line represents origins at each relative distance fit to a nonlinear regression curve with a 90% confidence interval (shading). $p$-values calculated using a one-sided, paired $t$-test between conditions for the first three relative distance bins (0, 0.01, 0.02). **d** Whole cell lysates of immortalized fallopian tube epithelial cells (FTEs) with doxycycline (DOX)-inducible BRCA2-

shRNA after 0, 3, and 6 days of treatment were analyzed by western blot for validation of knockdown (KD) and pRPA-related replication stress. Tubulin is a loading control for this blot and subsequent western blots. Results reproducible for at least 3 biological replicates. **e** (Right) Log$_2$ annotation enrichment of FTE Ok-seq origins at 0, 3, and 6 days of BRCA2-KD. $N$ = 2 biological Ok-seq replicates each (left). Zoomed TTS annotation (right); lines connect biological replicates. **f** Relative distance origin analysis of untreated and BRCA2-KD FTEs relative to TSSs of high-volume genes. For all subsequent experiments, 3 days of dox treatment was used for BRCA2-KD. The solid line represents origins at each relative distance fit to a nonlinear regression curve with a 90% confidence interval (shading). $p$-values calculated using a one-sided, paired $t$-test between conditions for the first three relative distance bins (0, 0.01, 0.02). **g** Relative distance origin analysis of untreated and BRCA2-KD FTEs relative to TTSs of high-volume genes. The solid line represents origins at each relative distance fit to a nonlinear regression curve with a 90% confidence interval (shading). $p$-values calculated using a one-sided, paired $t$-test between conditions for the first three relative distance bins (0, 0.01, 0.02). **h** Log$_{10}$ normalized gene FPKM and normalized gene length of candidate genes with new origins at TTSs with BRCA2-KD ($n$ = 418 genes). The dotted line represents the average FPKM and gene length of all genes ($N$ = 18250) from RNA-seq analysis of FTEs. **i** Two representative candidate genes with new origins at TTSs with BRCA2-KD in two Ok-seq biological replicates each with 0, 3, and 6 days of knockdown: *MYO1B* and *TLE3*. Source data are provided as a Source Data file.

phosphorylation of replication protein A (pRPA32-Ser33), a checkpoint signal for slowed/stalled replication forks. The level of pRPA32-Ser33 appears similar between 3 or 6 days of BRCA2-KD (Fig. 1d). To capture the more immediate effects on DNA replication dynamics upon BRCA2 loss, most assays henceforth were done using the 3-day time-point for DOX-induced BRCA2-KD. DOX treatment alone in shScramble FTEs had no impact on BRCA2 levels or phosphorylation of RPA (Supplementary Fig. 1a). We performed two biological replicates of Ok-seq and downstream origin analysis on FTEs with a time-course of BRCA2-KD. As expected, genomic annotation of replication origins in FTEs showed origins are most prevalent at TSSs, as well as a relative enrichment of intergenic origins, regardless of BRCA2 status (Fig. 1e). In contrast, other intragenic sites, such as exon, intron, and 3′UTR are underrepresented in both BRCA2-proficient and -deficient FTEs, consistent with previously published work that active transcription likely clears genes of MCM helicase complex to disfavor intragenic origin firing[56–59]. Intriguingly, we found that transient BRCA2-KD (both days 3 and 6 of DOX treatment) resulted in an increase in identified origins (dormant origins) that map to TTSs (Fig. 1e). Using the same relative distance analysis (as for HU treatment), we show that BRCA2-KD similarly caused an increase in origin usage oriented near TTSs of transcription high-volume genes in both Ok-seq replicates (Fig. 1g and Supplementary Fig. 2b). In contrast, BRCA2-KD did not significantly impact origin usage near TSSs (Fig. 1f and Supplementary Fig. 2a) by this origin call analysis. Importantly, this enrichment of origins at TTSs after BRCA2-KD was not observed in transcription low-volume genes (Supplementary Fig. 2c, d), when compared to all TSSs and TTSs (regardless of FPKM) (Supplementary Fig. 2e, f), or random genomic sites (Supplementary Fig. 2g), supporting that only origins at specific genes sets are affected by BRCA2-KD.

To gain more clarity on the orientation of these BRCA2-deficient with respect to high-volume gene TTSs, we separated origins calls from BRCA2-WT and BRCA2-KD into intragenic and intergenic origins (Supplementary Fig. 3a). Relative distance analysis reveals that specifically intergenic origins in BRCA2-KD are significantly enriched at high-volume TTSs compared to BRCA2-WT (Supplementary Fig. 3b), while intragenic origins in both BRCA2-WT and BRCA2-KD FTEs are oriented away from high-volume TTSs (Supplementary Fig. 3c). This is consistent with previously published work suggesting actively transcribed genes are cleared of loaded MCMs and active transcription might suppress origin firing[56–59]. Indeed, intragenic

origins in actively transcribed genes are not prevalent in FTEs, and BRCA2-KD did not impact levels of intragenic origin usage (Fig. 1e). However, the captured origins that do map to intragenic sites by our analysis (in both BRCA2-proficient and BRCA2-KD FTEs) occurred in low-expressing, long genes (Supplementary Fig. 3d–f). Thus, it is highly unlikely that replication stress causes new/dormant origin firing at intragenic sites of actively transcribed genes. Similarly, Ok-seq analysis of RPE-1 untreated and HU-treated cells reveals that intragenic origins only occur in low-expressing, long genes (Supplementary Fig. 3g, h).

Using an exact distance calculation between replication origins and TTS annotations, we identified candidate genes that have new replication origins near or downstream of TTSs in BRCA2-KD, not present in the relevant BRCA2-WT Ok-seq sample. 418 candidate genes were identified between the two Ok-seq replicates (Source Data). These candidate genes have higher-than-average expression and longer gene lengths when compared to all genes based on RNA-seq analysis of FTEs (Fig. 1h). Representative candidate genes with new TTS origins in BRCA2-KD samples are shown (Fig. 1i and Supplementary Fig. 4). Importantly, we also interrogated whether adjacent gene proximity impacts new (dormant) origin firing at candidate gene TTSs upon BRCA2-KD. It is conceivable that nearby origins from adjacent, active genes can influence the degree of dormant origin firing at the TTSs of candidate genes. To address this question, candidate genes with new origins at TTSs were computationally sub-divided into different categories: candidate genes that are nearest another gene (genes in tandem) whereby the downstream neighboring TSS is <50 kb away from the candidate gene TTS origin (Supplementary Fig. 3i, ii); convergent gene neighbors whereby the downstream neighboring TTS is <50 kb away from the candidate gene TTS origin (Supplementary Fig. 3i, iii); and candidate gene TTS origins that are far away from the nearest TSS or TTS of adjacent genes (>50 kb away) (Supplementary Fig. 3i, iv). Nearly 50% of all new origin firing at TTSs of candidate genes are far away from the nearest adjacent gene (either convergent or in tandem), and dormant origins fire the least frequently at TTSs of convergent gene neighbors (Supplementary Fig. 3j, left) compared to the distribution of all genes (Supplementary Fig. 3j, right). Representative candidate genes with new TTS origins in the context of neighboring gene orientation are shown (Supplementary Fig. 4a–c). Altogether, our Ok-seq analysis of FTEs shows increased dormant origin firing at TTSs of transcriptionally high-volume genes that are located away from nearby active genes.

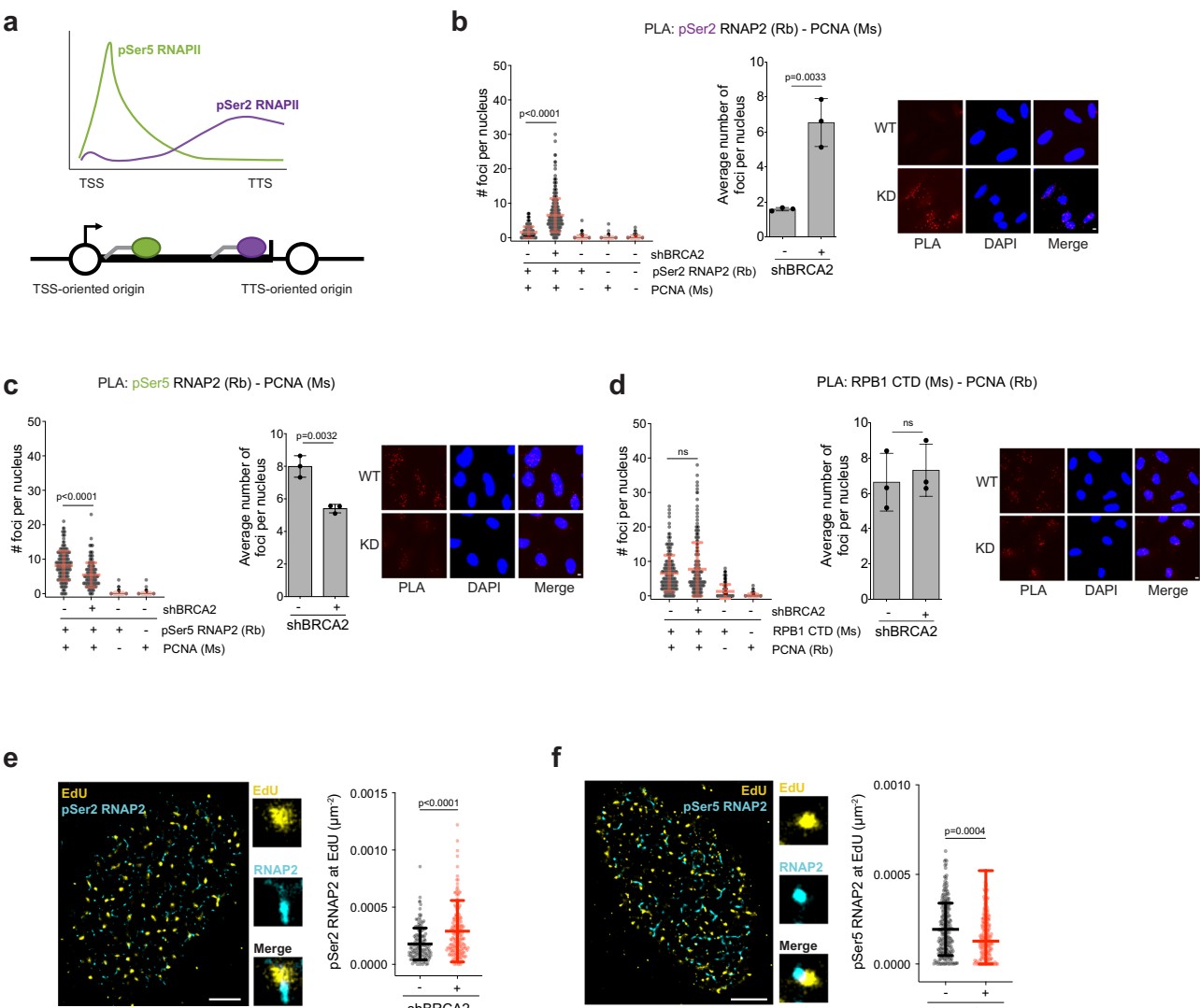

**Fig. 2 | BRCA2 knockdown causes HO-TRCs with elongating RNAP2. a** Schematic of pSer5- and pSer2-RNAP2 phospho-form abundance distributed across gene bodies relative to intergenic replication origin firing. **b** Proximity-ligation assay (PLA) of pSer2-RNAP2 and PCNA in WT- and KD-BRCA2 FTEs including single and no antibody controls. The data presented shows 300 nuclei from 3 biological replicates. *p*-values calculated using unpaired two-tailed *t*-tests. Left: number of foci per nucleus. Right: average number of foci per nucleus per biological replicates. Error bars = mean, std. Scale bar = 2 μm in representative pictures. **c** PLA of pSer5-RNAP2 and PCNA in WT- and KD-BRCA2 FTEs including single antibody controls. The data presented shows 200 nuclei from 3 biological replicates. *p*-values calculated using unpaired two-tailed *t*-tests. Left: number of foci per nucleus. Right: average number of foci per nucleus per biological replicates. Error bars = mean, std. Scale bar = 2 μm in representative images. **d** PLA of RPB1-CTD and PCNA in WT- and KD-BRCA2 FTEs including single antibody controls. The data presented shows 200 nuclei from 3

biological replicates. *p*-values calculated using unpaired two-tailed *t*-tests. Left: number of foci per nucleus. Right: average number of foci per nucleus per biological replicates. Error bars = mean, std. Scale bar = 2 μm in representative images. **e** Representative SMLM image shows EdU (yellow) and pSer2-RNAP2 (cyan). Scale bar = 2 μm. Scatter plot quantification measuring pSer2-RNAP2 at EdU in WT- and KD-BRCA2 (see "Methods" section). *p*-values calculated using unpaired two-tailed *t*-test from at least two biological replicates (WT: $N = 156$, KD: $N = 203$, where $N = \#$ of S-phase nuclei). Error bars = mean, std. **f** Representative SMLM image shows EdU (yellow) and pSer5-RNAP2 (cyan). Scale bar = 2 μm. Scatter plot quantification measuring pSer5-RNAP2 at EdU in WT- and KD-BRCA2. *p*-values calculated using unpaired two-tailed *t*-test from at least two biological replicates (WT: $N = 250$, KD: $N = 218$, where $N = \#$ of S-phase nuclei). Error bars = mean, std. Source data are provided as a Source Data file.

## BRCA2 deficiency elevates HO-TRCs at sites of elongating RNAP2

Replication stress can lead to conflicts (collisions) between replication forks and transcriptional types of machinery, known as transcription-replication conflicts (TRCs). Considering that BRCA2 deficiency leads to increased dormant origin firing at or downstream of TTSs of high-volume genes, we hypothesized that this aberrant origin firing could contribute to enhanced HO-TRCs. Different forms of RNAP2 are enriched at different regions of active genes, which may serve as reference markers for potential sites of TRCs. To test this, we performed proximity-ligation assays (PLA) for measuring the levels of the different functional forms of RNAP2 localized in close

proximity to active replication forks as a proxy for TRCs. Specifically, we used antibodies against two distinct phospho-forms of RNAP2, pSer5-RNAP2 and pSer2-RNAP2, corresponding to initiation and elongation transcriptional complexes, respectively (see schematic, Fig. 2a). Briefly, hypo- or unphosphorylated RNAP2 is loaded at TSSs, and is then phosphorylated on its repeated C-terminal domain (CTD) at serine-5 residues (pSer5) by CDK7/TFIIH. pSer5-RNAP2 is associated with transcription initiation and promoter-proximal pausing a few tens of bases downstream of TSSs[60]. Active transcriptional elongation is characterized by dephosphorylation of serine-5 residues[60] and phosphorylation of serine-2 residues (pSer2) by

CDK9/pTEFb[61]. Therefore, the pSer5-RNAP2 form peaks near the TSSs of active genes, whereas the pSer2-RNAP2 form accumulates along gene bodies, and is enriched at TTSs of active genes (Fig. 2a). Considering intragenic origin firing is suppressed in actively transcribed genes (Supplementary Fig. 3c–h), we propose that the proximity between various phospho-forms of RNAP2 and active forks arising from intergenic origins could mark distinct TSS- and TTS-oriented locations for TRC events (Fig. 2a). We found that the PLA signal between PCNA and pSer2-RNAP2 increased in BRCA2-KD FTEs, in comparison to no-DOX FTEs and no or single antibody controls (Fig. 2b). Increased pSer2-RNAP2 – PCNA PLA signal in BRCA2-KD was also observed specifically in S-phase (EdU+) nuclei (Supplementary Fig. 5a). Similar to pSer2-RNAP2, pThr4-RNAP2 is associated with transcription elongation and has been shown to be abundant at TTSs[60]. PLA interrogating interaction between pThr4-RNAP2 and PCNA shows increased signal with BRCA2-KD in S-phase (EdU+) nuclei (Supplementary Fig. 5b). Confirming the results of our DOX-inducible shBRCA2 FTEs, using two distinct siRNAs against BRCA2 in FTEs also showed increased PLA signal between pSer2-RNAP2 and PCNA (Supplementary Fig. 5c). Interestingly, in contrast to pSer2-RNAP2, pSer5-RNAP2 proximity to PCNA decreased in BRCA2-KD cells (Fig. 2c). PLA between an antibody that detects pan-RNAP2 (total) and PCNA failed to detect significant changes after BRCA2-KD (Fig. 2d), suggesting that only a subset of the total RNAP2 is engaging in TRCs.

We next performed single-molecule localization microscopy (SMLM)-STORM, coupled with automated data analysis routine[22,54,62–64], as an orthogonal approach to assess whether co-localization between the replication forks and transcriptional machinery is increased at the single-molecule level in BRCA2-KD FTEs. This approach is particularly advantageous for detecting low-level events and specific intermediates with high accuracy and has been broadly applied and validated in studies of replication, transcription, and DNA repair-associated processes, as described in detail in our prior work[22,54,62–64]. Briefly, SMLM is a powerful imaging technique that dramatically improves spatial resolution over standard, diffraction-limited microscopy techniques and can image biological structures at the molecular scale[65,66]. In SMLM, individual fluorescent molecules are computationally localized from diffraction-limited image sequences, and the localization of these fluorescent molecules is used to generate a super-resolution image, in order to define molecular structures/trajectories. To resolve replication forks and associated complexes in FTE cells, sample preparation, and SMLM imaging were performed as previously described[67], and resulting images were submitted to an automated analysis platform for quantifying specific molecular features in SMLM images[62,65–69]. FTEs were pulse-labeled for 10 min with EdU followed by click-chemistry fluorophore ligation and anti-phospho-RNAP2 antibody localization. Analysis of the levels of pSer2-RNAP2 signal associated with replication forks (represented by EdU signal) revealed a notable increase following depletion of BRCA2 (Fig. 2e), whereas the levels of pSer5-RNAP2 at nascent DNA showed a decrease (Fig. 2f). Collectively, our SMLM data are consistent with our PLA experimental results, and demonstrates that BRCA2 deficiency in FTEs leads to an increase in TTS-oriented TRCs, as measured by enhanced proximity/co-localization of the replication fork with elongating RNAP2 (pSer2-RNAP2). Based on the evidence that TRC events favor downstream regions of elongating RNAP2 rather than the initiating RNAP2, we suggest that forks generated from TTS origins converge with elongating RNAP2 to increase HO-TRCs at long, highly transcribed genes.

## CDC7 inhibition reduces both dormant origin firing and HO-TRCs in BRCA2-deficient cells

To determine the role(s) of the replication checkpoint response in regulating HO-TRCs in BRCA2-deficient cells, we tested the effects of several clinically relevant DNA replication checkpoint inhibitors and examined how they would influence HO-TRCs via alterations in replication-fork speed or origin firing efficiencies. Fork speed and origin firing density are inversely correlated to one another: conditions that decrease replication-fork speed will promote compensatory proximal dormant origin firing, whereas unrestrained fork progression (faster fork speed) will limit dormant origin firing through mechanisms of passive replication[70]. In unperturbed settings, the majority of constitutive origins that are tasked to replicate euchromatic regions are localized to TSSs of active genes, and efficient fork elongation across gene bodies prevents proximal dormant origin firing at TTSs (see schematics, Fig. 3a). Upon replication stress, characterized as reduced fork elongation rates (fork-slowing/stalling), dormant origins fire to rescue slowed/stalled proximal replication forks (Fig. 3a). The CDC7 kinase (DDK) complex phosphorylates the MCM helicase proteins to activate licensed origins[71]. It has been previously shown that inhibition of CDC7 activity limits the firing of new origins in the context of replication stress, despite having minimal effects on the firing from canonical origins[62,72,73] (Fig. 3a). Poly ADP-ribose polymerase 1 (PARP-1) is an enzyme that maintains genomic integrity through multiple mechanisms, including promoting fork-stalling and reversal of stressed forks, and preventing ssDNA gaps during DNA synthesis[74]. Inhibition of PARP activity with olaparib (PARPi) causes unrestrained fork progression (faster fork speed, despite the presence of ssDNA gaps behind the forks)[75,76], thereby overcoming the need for dormant origin firing to help compensate for stalled forks (Fig. 3a). Finally, the ATR kinase is a master regulator of replication stress, in part, by coordinating early- and late-firing origins of the genome to ensure timely DNA replication and to prevent RPA exhaustion[72,77]. It was shown that ATR inhibition (ATRi, VE-821) causes unscheduled origin firing and concomitant fork-slowing (Fig. 3a)[78]. Therefore, we utilized these three generic replication stress-modulating tools to study whether dormant origin firing perturbations can influence HO-TRC events in BRCA2-deficient cells.

To interrogate replication-fork speed and origin density, we performed single-molecule DNA fiber analysis as previously described[65]. Briefly, cycling cells are sequentially pulse-labeled with two nucleotide analogs (IdU, CldU) for 20 min each, followed by cell lysis and spreading of DNA onto slides. Slides are stained with fluorescent antibodies and elongating DNA tracks are measured as second-label track length (Fig. 3b). DNA fiber analysis of shBRCA2 FTEs revealed that loss of BRCA2 causes impaired fork elongation (Fig. 3b). CDC7i treatment resulted in increased fork speed in control FTEs, but not in BRCA2-deficient FTEs (Fig. 3b). DNA fiber analysis was also used to determine origin density through measurement of inter-origin distance between the center of adjacent first-label nucleotide analogs (Fig. 3c). BRCA2 loss in FTEs caused a decrease in inter-origin distance indicating increased origin density, consistent with Ok-seq analysis of elevated dormant origins at TTSs of high-volume genes (Fig. 1i). CDC7i treatment decreased origin density in both control and BRCA2-deficient FTEs. We showed that decreased origin density caused by CDC7i in BRCA2-deficient cells can occur without a compensatory increase in fork speed (Fig. 3b, c). Importantly, PLA analysis of pSer2-RNAP2 and PCNA in BRCA2-deficient FTEs treated with CDC7i revealed that CDC7 kinase inhibition can rescue or prevent HO-TRCs (Fig. 3d and Supplementary Fig. 5e). This suggests that dormant origin firing is the major culprit of HO-TRCs in BRCA2-deficient cells. In contrast, PLA analysis of pSer5-RNAP2 and PCNA co-localization in BRCA2-deficient FTEs treated with CDC7i showed little to no impact (Fig. 3e and Supplementary Fig. 5f). This implies that potentially canonical forks originating at TSSs are not impacted by CDC7i treatment in BRCA2-deficient cells, and the rescue phenotype of the PLA signal between pSer2-RNAP2 and PCNA is not simply due to a global decrease in DNA synthesis levels (we elaborate on the latter point below).

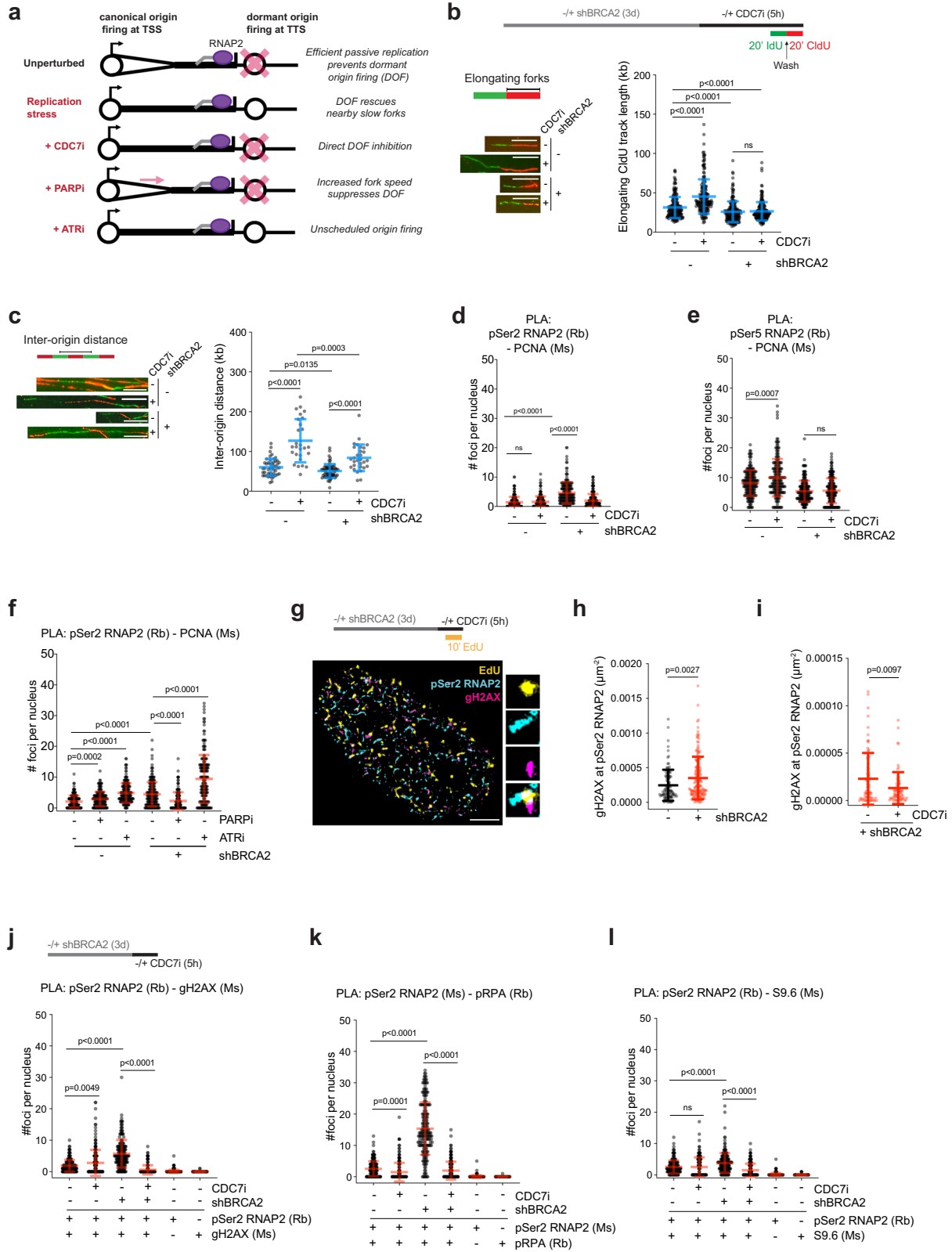

To determine the effects of PARPi and ATRi treatment on HO-TRCs, we first validated their effects on replication-fork speed and origin density in BRCA2-deficient FTEs. As expected, DNA fiber analysis of FTEs treated with olaparib showed increased fork speed in both BRCA2-proficient and BRCA2-KD FTEs (Supplementary Fig. 5g). Conversely, ATRi in FTEs shows decreased fork speed independent of BRCA2 status (Supplementary Fig. 5g).

Concordantly, analysis of inter-origin distance by DNA fiber analysis revealed that PARPi decreases origin density in both BRCA2-proficient and -KD FTEs, while ATRi increases origin density regardless of BRCA2 status (Supplementary Fig. 5h). To determine the impact of these replication perturbations on HO-TRCs, we again performed PLA analysis of pSer2-RNAP2 and PCNA, which revealed that PARPi treatment can rescue HO-TRCs in BRCA2-deficient FTEs,

**Fig. 3 | Replication stress modulators alter dormant origin firing and HO-TRCs in BRCA2-deficient cells. a** Schematic showing the impact of replication stress modulators, CDC7i, PARPi, and ATRi, on fork speed and dormant origins. In unperturbed cells, passive replication prevents the firing of nearby licensed origins. Under-replication stress, failure of fork progression causes the nearby firing of dormant origins[54]. CDC7i directly inhibits dormant origins under-replication stress[73]. PARPi increases fork speed[76]. ATRi causes unscheduled origin firing[78]. **b** DNA fiber analysis of sequentially labeled nucleotide analogs (IdU and CldU, each 20′) followed by DNA spreading and fluorescent labeling to measure fork elongation as CldU track length in kilobases in FTEs. Representative images are shown of DNA fibers from WT- and KD-BRCA2 FTEs ± 5 μM CDC7i, 5 h. Scale bars = 10 μm. The scatter plot shows the quantification of elongating CldU track length. *p*-values calculated using unpaired two-tailed *t*-tests of 200 fibers per condition from three biological replicates. Error bars = mean, std. **c** DNA fiber analysis to measure inter-origin distance in kilobases. Representative images of DNA fibers from WT- and KD-BRCA2 ± CDC7i. Scale bars = 10 μm. The scatter plot shows the quantification of inter-origin distance. *p*-values calculated using unpaired two-tailed *t*-tests from three biological replicates: WT-BRCA2(−)CDC7i, *N* = 48; WT-BRCA2(+) CDC7i, *N* = 27; KD-BRCA2(−)CDC7i, *N* = 65; KD-BRCA2(+)CDC7i, *N* = 30, where *N* = # of IOD fibers analyzed. Error bars = mean, std. **d** PLA of pSer2-RNAP2 and PCNA in WT- and KD-BRCA2 ± CDC7i. The data presented shows 200 nuclei from 3 biological replicates. Number of foci per nucleus, *p*-values calculated using unpaired two-tailed *t*-tests. Error bars = mean, std. **e** PLA of pSer5-RNAP2 and PCNA in WT- and KD-BRCA2 ± CDC7i. The data presented shows 200 nuclei from 3 biological replicates. Number of foci per nucleus, *p*-values calculated using unpaired two-tailed *t*-tests. Error bars = mean, std. **f** PLA of pSer2-RNAP2 and PCNA in WT- and KD-BRCA2 ± PARPi or ATRi. *p*-values calculated using unpaired two-tailed *t*-tests of 200 nuclei per condition from 3 biological replicates. Error bars = mean, std. **g** Representative SMLM image of EdU pulse-labeled FTE nucleus stained with pSer2-RNAP2 and γH2AX. Scale bar = 2 μm. **h** Scatter plot quantification measuring SMLM of gH2AX at pSer2-RNAP2 in WT- and KD-BRCA2. *p*-values calculated using unpaired two-tailed *t*-test of at least two biological replicates. WT = 108; KD, *N* = 165, where *N* = # of S-phase nuclei. Error bars = mean, std. **i** Scatter plot quantification measuring SMLM of gH2AX at pSer2-RNAP2 in KD-BRCA2 ± CDC7i. *p*-values calculated using unpaired two-tailed *t*-test of at least two biological replicates. (−) CDC7i, *N* = 80; (+)CDC7i, *N* = 73, where *N* = # of S-phase nuclei. Error bars = mean, std. **j** PLA of pSer2-RNAP2 and gH2AX in WT- and KD-BRCA2 ± CDC7i including single antibody controls. *p*-values calculated using unpaired two-tailed *t*-tests of 200 nuclei per condition from 3 biological replicates. Error bars = mean, std. **k** PLA of pSer2-RNAP2 and pRPA-Ser33 in WT- and KD-BRCA2 ± CDC7i including single antibody controls. *p*-values calculated using unpaired two-tailed *t*-tests of 200 nuclei per condition from 3 biological replicates. Error bars = mean, std. **l** PLA of pSer2-RNAP2 and S9.6 in WT- and KD-BRCA2 ± CDC7i including single antibody controls. *p*-values calculated using unpaired two-tailed *t*-tests of 200 nuclei per condition from 3 biological replicates. Error bars = mean, std. Source data are provided as a Source Data file.

while ATRi exacerbates HO-TRCs regardless of BRCA2 status (Fig. 3f and Supplementary Fig. 5i). Interestingly, in BRCA2-proficient and -deficient FTEs, both PARPi and ATRi treatments showed increased PLA signal between pSer5-RNAP2 and PCNA, suggesting that increased fork speed (albeit with replication gaps), in the case of PARPi, or unscheduled origin firing at canonical replication origins (at TSSs) or in gene bodies, in the case of ATRi, exacerbates both pSer5- and pSer2-RNAP2-associated TRCs (Supplementary Fig. 5j). Collectively, these results affirm the impact of fork-stalling and aberrant origin firing on HO-TRCs occurring near the TTS of active genes in BRCA2-deficient cells.

## HO-TRCs are potential hotspots for genomic instability in BRCA2-deficient ovarian cancer precursor cells

FTE-derived HGSOCs are characterized as having elevated levels of intrinsic DNA damage[3,5]. To determine whether HO-TRCs in BRCA2-deficient FTEs cause DNA damage, we used SMLM probing for γH2AX (a marker of DNA breaks) that are localized at pSer2-RNAP2 in S-phase nuclei (Fig. 3g). BRCA2-KD in FTEs led to an increase in the level of γH2AX signal at elongating pSer2-RNAP2 in S-phase FTE compared to untreated (Fig. 3h), suggesting that HO-TRCs at TTS contribute to genomic instability. Intriguingly, CDC7i treatment decreased γH2AX signal at elongating pSer2-RNAP2 in S-phase BRCA2-KD FTEs (Fig. 3i). PLA analysis between pSer2-RNAP2 and gH2AX also showed increased DNA damage at HO-TRCs in BRCA2-KD FTEs, which was abrogated upon CDC7i treatment, consistent with the SMLM results (Fig. 3j and Supplementary Fig. 6a). To rule out any potential off-target effects of the CDC7i utilized in our study, we examined the nuclear EdU density as well as the total pSer2-RNAP2 levels in cells to determine the global impact of CDC7i treatment on DNA synthesis and transcription levels in BRCA2-WT and -KD cells. These results show that CDC7i does not significantly impact overall EdU-incorporated nascent synthesis by SMLM (Supplementary Fig. 6b), consistent with previously published findings from our group[62]. CDC7i decreases global pSer2-RNAP2 in BRCA2-WT cells, likely due to simultaneous off-target inhibition of CDK9 by this drug[79] (Supplementary Fig. 6c). Loss of BRCA2 in FTEs decreases global nuclear pSer2-RNAP2 abundance by STORM, however, CDC7i does not further decrease global pSer2 levels in BRCA2-deficient cells (Supplementary Fig. 6c), suggesting the decreased HO-TRCs observed by PLA in CDC7i treatment (Fig. 3d), and the decreased transcription-associated DNA damage associated with CDC7i

(Fig. 3l, j) is not simply due to CDC7i impacting total elongating RNAP2, nor global DNA synthesis.

Previous studies have linked HO-TRCs to be sites of phospho-RPA-enriched replication stress and R-loop accumulation[80]. Consistent with a previous study that finds TTSs of transcriptionally active genes are sites of pSer33-RPA by ChIP-seq[80], we found that loss of BRCA2 led to increased PLA signal between pSer2-RNAP2 and pSer33-RPA, which can be rescued by CDC7i treatment (Fig. 3k and Supplementary Fig. 6d). Finally, HO-TRCs are associated with persistent, toxic R-loop formation in contrast to CD-TRCs in certain models[36], and loss of BRCA2 causes both transcriptional dysregulation and R-loop persistence[23]. Here, we showed that BRCA2-deficient FTEs have increased R-loops (as marked by S9.6 RNA-DNA hybrid antibody) associated with pSer2-RNAP2 by PLA analysis (Fig. 3l and Supplementary Fig. 6e), which was also rescued by CDC7i. Altogether, these results demonstrate that it is likely that elevated HO-TRCs in BRCA2-deficient cells are sites of genomic instability caused by proximal dormant origin firing events.

## BRCA2 and RNase H2 cooperate to mitigate HO-TRCs at transcription termination sites

It was previously reported that BRCA2 associates with RNAP2 to promote transcriptional elongation and to prevent R-loop accumulation at transcriptionally active genes[23]. To determine whether BRCA2 is recruited directly to HO-TRCs, we treated FTE cells with low-dose (0.2 mM) hydroxyurea (HU), which causes fork slowdown and induces dormant origin firing[53,54]. By PLA, we found that low-dose HU induces HO-TRCs between pSer2-RNAP2 and PCNA (Supplementary Fig. 5k), consistent with dormant origin firing at TTSs of high-volume genes (Fig. 1c). We then performed SMLM on untreated and HU-treated FTEs, pulse-labeled with EdU, to determine whether BRCA2 responds directly to TRCs induced by dormant origins (Fig. 4a). These measurements showed that replication stress due to HU treatment leads to an increase in the levels of both pSer2-RNAP2 and BRCA2 molecules at replication forks (EdU signal) (Fig. 4b). These results provide evidence that BRCA2 is recruited to HO-TRCs upon HU-mediated dormant origin firing.

To determine the extent to which BRCA2 recruitment to HO-TRCs caused by HU-induced dormant origin firing relies on active transcription elongation, FTEs were treated with low-dose HU, followed by treatment with 5/6-Dichlorobenzimidazole 1-β-D-ribofuranoside (DRB), an inhibitor of transcriptional elongation. Consistent with our

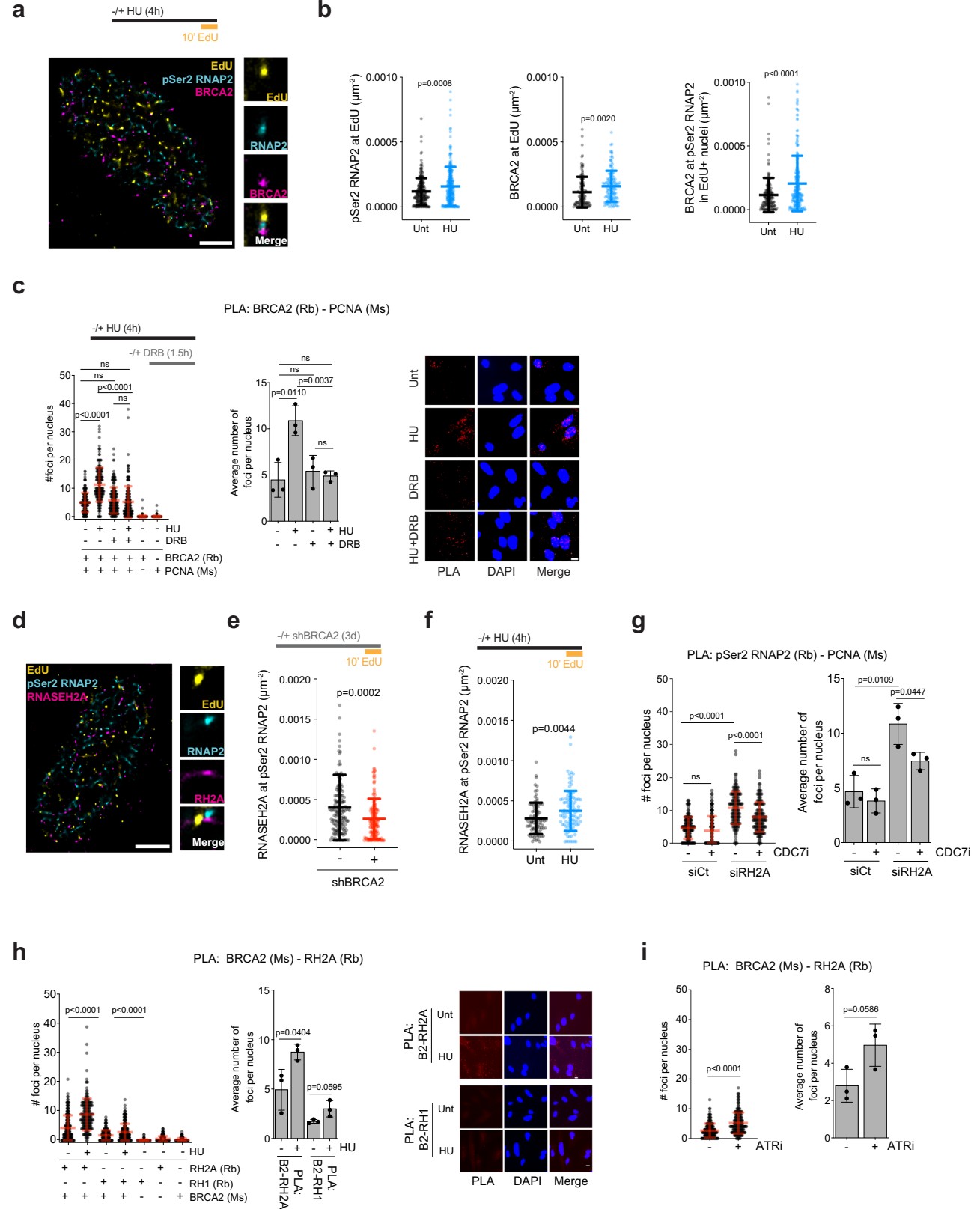

SMLM results, the PLA signal between BRCA2 and PCNA was increased upon HU treatment, revealing the elevated recruitment of BRCA2 to active forks, in comparison to untreated or single antibody controls (Fig. 4c). DRB treatment alone did not alter BRCA2 recruitment to forks, but it suppressed BRCA2 recruitment to forks after HU treatment (Fig. 4c). These results suggest that recruitment of BRCA2 to the

replication fork, in part, requires transcription elongation in response to replication stress.

A recent report implicated RNase H2 in RNAP2-directed transcriptional regulation and R-loop resolution[21]. RNase H2 has also been previously shown to interact with BRCA2 to prevent DNA damage-associated R-loops enriched in S-phase[22]. Therefore, we hypothesized

**Fig. 4 | BRCA2 and RNase H2 cooperate together to mitigate HO-TRCs.**
**a** Representative SMLM images of EdU pulse-labeled FTEs ± 0.2 mM HU treatment (4 h) stained with pSer2-RNAP2 and BRCA2. Scale bar = 2 μm. **b** Scatter plot quantification measuring pSer2-RNAP2 at EdU in untreated and HU-treated FTEs (left; untreated, $N = 234$; HU, $N = 255$, where $N$ = # of S-phase nuclei), BRCA2 at EdU (middle; untreated, $N = 125$; HU, $N = 139$), and BRCA2 at pSer2-RNAP2 (right; untreated, $N = 162$; HU, $N = 219$). $p$-values calculated using unpaired two-tailed $t$-tests of at least two biological replicates. Error bars = mean, std. **c** PLA of BRCA2 and PCNA in FTEs treated with 0.2 mM HU for 4 h, and/or 100 μM DRB for 90 min, including single antibody controls. The data presented shows 200 nuclei from 3 biological replicates. $p$-values calculated using unpaired two-tailed $t$-tests. Left: number of foci per nucleus. Right: average number of foci per nucleus per biological replicates. Error bars = mean, std. Scale bar = 10 μm in representative images. **d** Representative SMLM image of EdU pulse-labeled FTE cell stained with pSer2-RNAP2 and RNASEH2A. Scale bar = 2 μm. **e** Scatter plot quantification measuring RNASEH2A at pSer2-RNAP2 in EdU-labeled WT- and KD-BRCA2. $p$-values calculated using unpaired two-tailed $t$-test of at least two biological replicates. Unt, WT = 160; KD, $N = 169$, where $N$ = # of S-phase nuclei. Error bars = mean, std. **f** Scatter plot

quantification measuring RNASEH2A at pSer2-RNAP2 in EdU-labeled untreated and HU-treated FTEs. $p$-values calculated using unpaired two-tailed $t$-test of at least two biological replicates. Unt, $N = 92$; HU, $N = 101$. Error bars = mean, std. **g** PLA of pSer2-RNAP2 and PCNA in siControl or siRNASEH2A FTEs ± CDC7i. The data presented shows 150 nuclei from 3 biological replicates. $p$-values calculated using unpaired two-tailed $t$-tests. Left: number of foci per nucleus. Right: average number of foci per nucleus per biological replicates. Error bars = mean, std. **h** PLA of BRCA2 and RNASEH2A or BRCA2 and RNase H1 in FTEs ± 0.2 mM HU for 4 h, including single antibody controls for each PLA. The data presented shows 200 nuclei from 3 biological replicates. $p$-values calculated using unpaired two-tailed $t$-tests between conditions. Left: number of foci per nucleus. Right: average number of foci per nucleus per biological replicates. Error bars = mean, std. Scale bar = 10 μm in representative pictures. **i** PLA of BRCA2 and RNaseH2A or BRCA2 in FTEs ± ATRi. The data presented shows 200 nuclei from 3 biological replicates. $p$-values calculated using unpaired two-tailed $t$-tests between conditions. Left: number of foci per nucleus. Right: average number of foci per nucleus per biological replicates. Error bars = mean, std. Source data are provided as a Source Data file.

that RNase H2 cooperates with BRCA2 to prevent HO-TRCs. We performed SMLM on FTE cells pulse-labeled with EdU and probed with antibodies against pSer2-RNAP2 and RNASEH2A, the catalytic subunit of RNase H2 (Fig. 4d). We found that the loss of BRCA2 compromised RNASEH2A recruitment to pSer2-RNAP2 in S-phase cells (Fig. 4e). We next interrogated whether RNase H2 is directly recruited to TRCs, similar to BRCA2 (Fig. 4b). We observed that HU-treated cells show increase levels of RNASEH2A associated with pSer2-RNAP2 in EdU-positive cells when compared to untreated cells (Fig. 4f). Knockdown of RNASEH2A causes increased TRCs with elongating RNAP2 enriched at TTSs, shown by PLA analysis of pSer2-RNAP2 and PCNA (Fig. 4g). TTS-oriented TRCs in RNASEH2A-deficient FTEs are, to an extent, dependent on dormant origin firing (effect on CDC7i). Finally, FTEs treated with low-dose of HU have increased PLA signal between BRCA2 and RNASEH2A compared to untreated FTEs or single antibody controls (Fig. 4h), suggesting BRCA2 and RNase H2 interaction is increased in a replication stress-dependent manner. ATRi causes aberrant origin firing and TRCs with elongating RNAP2 in BRCA2-proficient FTEs (Supplementary Fig. 5g and Fig. 3f, respectively). Similar to FTEs treated with HU, PLA between BRCA2 and RNASEH2A shows an increased signal upon ATRi treatment. Interestingly, BRCA2 and RNase H1 also show increased interaction by PLA upon low-dose HU treatment (Fig. 4h). RNase H1 and RNase H2 both resolve cotranscriptional R-loops through the degradation of the RNA moiety of RNA:DNA hybrids[81]. Collectively, these results demonstrate that BRCA2 and RNase H2 functionally cooperate to suppress HO-TRCs, potentially through mitigation of toxic R-loop accumulation at these sites.

## R-loops accumulate in BRCA2-deficient cells at HO-TRC-prone genes

Stabilized R-loops are associated with HO-TRCs in a plasmid-based reporter system[36], and while both RNases H1 and H2 degrade the RNA moiety of RNA:DNA hybrids and R-loops, overexpression of single subunit RNase H1 has been used extensively to study persistent R-loops in human cell models[81,82]. To determine whether the disruption of R-loop homeostasis contributes to HO-TRCs in BRCA2-deficient cells, we transiently overexpressed V5-tagged human RNase H1 in FTEs as a way to broadly reduce R-loops in cells (Fig. 5a). RNase H1 overexpression reduces HO-TRCs (as measured by PLA between pSer2-RNAP2 and PCNA) in BRCA2-KD cells (Fig. 5b), suggesting that R-loop persistence in BRCA2-KD conditions contributes to HO-TRCs. We suspected our candidate genes identified by Ok-seq with new origins at the TTSs in BRCA2-KD are regions of increased persistent R-loops, consistent with our finding that HO-TRCs caused by BRCA2 deficiency specifically involving pSer2-RNAP2 have elevated R-loops compared to BRCA2-WT (Fig. 3l). To explore this hypothesis, we performed

DNA:RNA hybrid immunoprecipitation (DRIP)-qPCR analysis of TTS of candidate genes in FTEs transfected with empty vector control or RNase H1 in BRCA2-WT or -KD cells. DRIP-qPCR analysis reveals that BRCA2 loss causes increased R-loops at six candidate genes, the increased signal of which is rescued by RNase H1 overexpression (Fig. 5c). Interrogating three candidate genes further, we find by DRIP-qPCR that there is not an increased R-loop signal in BRCA2-KD FTEs at either the TSS or an intron of each of these genes (Supplementary Fig. 7a). Importantly, two low transcription, long genes with new origins at the TTS in BRCA2-KD show no increased R-loop signal at their TTS in BRCA2-KD (Supplementary Fig. 7b), suggesting active transcription is necessary for persistent R-loops caused by HO-TRCs. Collectively, these results suggest that BRCA2-deficient dormant origin firing causes local persistent R-loops at TTSs of long, highly transcribed genes.

Using Ok-seq analysis to identify unique replication origins in BRCA2-KD FTEs revealed new origins (dormant origins) firing specifically at transcriptionally high-volume genes (Fig. 1). Based on our results, we hypothesized that all dividing cells experiencing replication stress, regardless of cell type, are prone to harbor HO-TRCs and genomic instability at highly transcribed, long genes. To validate our prediction, we performed RNA-seq analysis of FTEs, and after using the hallmark gene set enrichment analysis (GSEA), we found that long (gene length >50 kb) and highly transcribed genes (top 500 genes based on FPKM) are enriched for multiple oncogenic pathways, including epithelial-to-mesenchymal transition (EMT), mTORC1 signaling, and angiogenesis (Fig. 5d). Intriguingly, using Ok-seq to curate candidate genes with new origins at TTSs in BRCA2-deficient FTEs, we similarly discovered that these identified genes are also associated with several similar oncogenic pathways including EMT (Fig. 5e). Thus, HO-TRCs occurring at these specific genic regions may be an important contributor to somatic sequence alterations that drive early FTE oncogenesis. To understand the clinical relevance of this BRCA2 mutant phenotype, we turned to earlier studies characterizing somatic sequence alterations in lesions of mutant *BRCA2* (*mutBRCA2*) patient samples. Labidi-Galy et al. previously analyzed whole-exome sequencing (WES) data of nine patients with serous tubal intra-epithelial carcinoma (STIC) lesions or HGSOCs[2]. Two of these patients had germline *BRCA2* mutations and underwent prophylactic bilateral salpingo-oophorectomy. Intriguingly, GSEA of genes with somatic sequence alterations (SSAs) in the STIC lesion of one *mutBRCA2* patient revealed hallmark gene enrichment for EMT (Supplementary Fig. 7c). Interestingly, there was no other hallmark gene enrichment for other pathways in any other patient sample. Next, genes with sequence alterations from six *mutBRCA2* HGSOC patient samples from TCGA (Firehose) were compared to all HGSOC patient samples with mutant

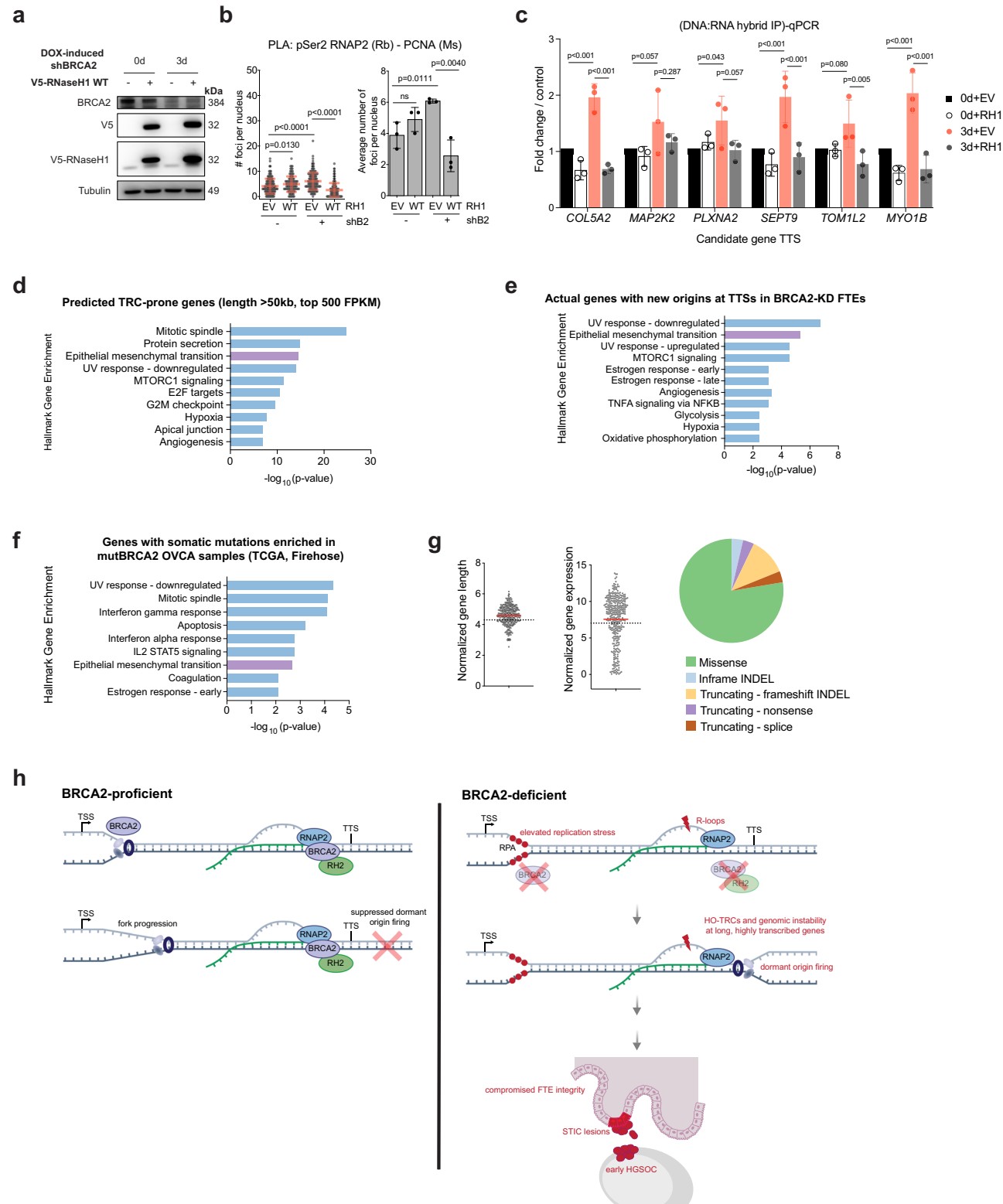

TP53 (mutTP53), as mutTP53 is ubiquitous in early HGSOC pathogenesis[2,5]. GSEA showed that genes with mutations enriched in mutBRCA2 samples compared to those in mutTP53 samples are similarly enriched for oncogenic pathways, including EMT (Fig. 5f). These mutated genes enriched in mutBRCA2 samples are also longer, and more highly expressed than average, based on RNA-seq analysis (Fig. 5g); consistent with the trend of HO-TRC-prone genes in BRCA2-deficient FTEs by Ok-seq (Fig. 1h). Further characterization reveals

these mutations are mostly missense mutations, followed by truncating mutations (frameshift INDELs, nonsense, or splice mutations), and lastly inframe INDELs (Fig. 5g). Finally, to determine whether there is a meaningful correlation between predicted HO-TRC-prone genes and dysfunctional protein activity that could result in EMT phenotypes in early HGSOC oncogenesis, we explored mutBRCA2 proteome data from the Clinical Proteomic Tumor Analysis Consortium (McDermott et al., CPTAC)[83]. KEGG pathway analysis of MS-proteomic data from a

**Fig. 5 | HO-TRCs are sites of genomic instability in BRCA2-deficient cells. a** Whole cell lysates of FTEs were analyzed by western blot. Empty vector or V5-tagged RNase H1 (RH1) plasmids were transiently transfected for 24 h prior to collection in WT- or KD-BRCA2 FTEs. Results reproducible for at least 3 biological replicates. **b** PLA of pSer2-RNAP2 and PCNA of WT- and KD-BRCA2 ± RH1 over-expression. The data presented shows at least 150 nuclei from 3 biological replicates. *p*-values calculated using unpaired two-tailed *t*-tests between conditions. Left: number of foci per nucleus. Right: average number of foci per nucleus per biological replicates. Error bars = mean, std. **c** DRIP-qPCR analysis of 6 candidate gene TTSs in WT- and KD-BRCA2 FTEs ± RH1 overexpression. The bar graph shows % of input normalized to the control sample (0d + EV) of 3 biological replicates. *p*-values calculated using two-way ANOVA with Tukey's multiple comparisons test. Error bars = mean, std. **d** Hallmark gene set analysis (GSEA) of genes length >50 kb and highly expressed (top 500 FPKM) from RNA-seq of FTEs. Top hallmark gene sets are shown as -log$_{10}$(p-value) *p*-values calculated using the hypergeometric distribution of overlapping genes over all genes in the gene universe. **e** Hallmark GSEA of candidate genes with new origins at TTSs in shBRCA2-KD based on Ok-seq

origin analysis. *p*-values calculated using the hypergeometric distribution of overlapping genes over all genes in the gene universe. **f** Hallmark GSEA of genes with mutations enriched in mut*BRCA2* samples vs all mut*TP53* samples in ovarian serous cystadenocarcinoma patients (TCGA Firehose study). *p*-values calculated using the hypergeometric distribution of overlapping genes over all genes in the gene universe. **g** Characterization of genes with mutations enriched in mut*BRCA2* samples vs all mut*TP53* samples in ovarian serous cystadenocarcinoma patients (*N* = 376) (TCGA Firehose study): log normalized gene length (left); log normalized gene expression (RNA-seq V2 RSEM) (middle); mutation type (right). **h** Schematic model of HO-TRCs at TTSs in an early BRCA2-deficient HGSOC. Left: BRCA2 protects stalled replication forks to promote fork progression of canonical forks at TSSs. BRCA2 promotes transcription elongation and R-loop resolution through RNase H2 recruitment. Right: in BRCA2 deficiency, increased replication stress and transcriptional dysregulation. Dormant origin firing at TTSs of long, highly transcribed genes causes HO-TRCs with elongating RNAP2 and DNA damage. TRC-prone genes maintain fallopian tube epithelial (FTE) integrity, which is compromised in early BRCA2-deficient HGSOC lesions. Source data are provided as a Source Data file.

mut*BRCA2* patient revealed that the most abundant proteins (log$_2$ fold change >1.5) in the normal FTE samples were enriched for pathways of focal adhesion and extracellular matrix-receptor interactions (Supplementary Fig. 7d). Conversely, the least abundant proteins (log$_2$ fold change <−1.5) in the HGSOC sample from the same patient were enriched in these same pathways, suggesting that decreased expression of proteins involved in the maintenance of epithelial integrity contribute to early FTE-derived HGSOC oncogenesis.

Here, we provide a schematic model describing our study on the role of BRCA2 in preventing HO-TRCs and genomic instability near TTSs, which is primarily driven by replication stress-induced dormant origin firing. In wild-type cells, BRCA2 prevents replication stress by maintaining replication-fork integrity likely through fork protection mechanisms (i.e. preventing fork reversals and nuclease degradation of nascent DNA) (Fig. 5h)[7–11]. Efficient fork progression (normal fork speed) suppresses downstream dormant origin firing through passive replication mechanisms. BRCA2 also promotes transcriptional elongation alongside RNase H2 to help prevent spurious R-loop accumulation[21,23]. However, in BRCA2-deficient cells, canonical forks originating at TSSs of transcriptionally active genes can experience fork slowdown or -stalling. Replication stress at these TSS-oriented canonical forks will lead to downstream dormant origin firing, rendering these genes more prone to HO-TRCs and enhanced genomic instability. R-loops persist at HO-TRCs due to the inability of BRCA2 to recruit RNase H2 to these sites. In the FTEs, we propose that HO-TRC-prone genes prevent oncogenesis through multiple mechanisms, including the maintenance of epithelial cell-type integrity. DNA mutations and transcriptional, and post-transcriptional dysregulation in these genes may compromise the FTE integrity, leading to early HGSOC oncogenesis when coupled to *mutTP53*.

## Discussion

Our findings support two interrelated mechanisms by which BRCA2 prevents HO-TRCs: (1) mitigation of dormant origin firing through the maintenance of replication-fork stability, and (2) prevention of persistent replication-stress-induced R-loop accumulation through RNase H2 recruitment. We propose that canonical forks originating from TSSs of transcriptionally high-volume genes are more inclined to slow down or stall in BRCA2-deficient cells, thereby inciting downstream dormant origin firing at TTSs of these genes (Fig. 1). Our Ok-seq origin call analysis described herein reveals that intragenic origin firing is uncommon, such that neither HU-mediated replication stress nor BRCA2 loss cause intragenic origin firing in highly transcribed genes. However, in general, the origins that do map within genic sites are occurring in low-expressing or transcriptionally silent genes. These results are consistent with recent work revealing that loaded MCM helicase complexes are likely

cleared from gene bodies by elongating RNAP2[57,59,84,85]. Therefore, our Ok-seq analysis of canonical and dormant origins provides a snapshot of the replication stress response to immediate BRCA2 loss, revealing that dormant origins fire from MCM helicase complexes redistributed at or just downstream of the termini of long and highly transcribed genes in BRCA2-deficient cells. Based on our findings, dormant origin firing in both BRCA2-KD and low-dose HU-treatment conditions leads to HO-TRCs at TTSs of active, long genes. Several targeted therapeutics that are known to perturb and modulate replication-fork speed/stalling and origin firing (e.g. CDC7i, PARPi, ATRi) can modulate HO-TRCs in BRCA2-deficient cells, in part, through CDC7-mediated dormant origin firing. Intriguingly, both PLA and SMLM techniques revealed that BRCA2 loss led to the decreased association between pSer5-RNAP2 and PCNA. One explanation for this could be that BRCA2 deficiency causes a decrease in canonical origin usage. Because origins localize immediately upstream of TSSs of highly transcribed genes (Ok-seq analysis), these interactions captured by PLA of pSer5-RNAP2 and PCNA likely represent co-directional collisions (CD-TRCs) between the nascent replication forks and RNAP2 from the transcription-initiating complexes. Conditions that increase fork speed (PARPi) or global asynchronous origin firing (ATRi) also cause an increase in pSer5-RNAP2-PCNA PLA in both BRCA2-proficient and -deficient cells, suggesting canonical origin/fork usage is modulated, and detectable by PLA as well. Therefore, our multi-pronged approach, using Ok-seq, PLA, and super-resolution microscopy tools, allowed us to map HO-TRCs genome-wide across gene bodies of highly transcribed genes. Collectively, our results revealed a new mechanism by which BRCA2 can directly mitigate toxic R-loops at HO-TRCs through the recruitment of RNase H2, and indirectly suppress HO-TRCs through the passive inhibition of replication stress-induced dormant origin firing. This provides a unique insight into the relationships between fork speed, origin firing, and the propensity for HO-TRCs to occur across gene bodies in various genetic and therapeutic perturbations.

It has been reported that BRCA2 directly binds RNAP2 to promote transcription elongation and prevent R-loop accumulation[23,24]. Persistent R-loops and R-loop-associated genomic instability are associated with HO-TRCs[16,36,37,86]; the former is mitigated in one study by direct RNase H2 recruitment to RNAP2 at transcriptionally active genes[21]. Another study found a role for BRCA2 in RNase H2 recruitment to DNA damage sites[22]. Our results show that BRCA2, alongside RNase H2, directly responds to HO-TRCs. Activation of dormant origins by low-dose HU treatment also causes HO-TRCs (as measured by PLA and SMLM analysis). Importantly, under these conditions, RNase H2 recruitment to HO-TRCs is dependent on BRCA2. Thus, the BRCA2-RNase H2 axis may provide a critical function in R-loop mitigation through transcriptional elongation processes during normal S-phase

progression, as defective transcription elongation is linked to persistent R-loops[87]. We speculate that nascent RNA synthesis is somewhat decreased in acute BRCA2-KD, similar to RNASEH2A-KD[21], and both BRCA2 and RNase H2 have been shown to directly associated with RNAP2[21,23]. Therefore, we propose that BRCA2 and RNase H2 play a critical role in promoting efficient, error-free transcription elongation and R-loop resolution during S-phase at highly transcribed and long genes during DNA replication.

Previous studies have linked HO-TRCs to be associated with regions of persistent R-loops and replication stress, in contrast to CD-TRCs[36]. Indeed, we discovered that endogenous sites of HO-TRCs are enriched for gH2AX, pRPA, and R-loop accumulation, which can be mitigated by limiting dormant origin firing with CDC7i treatment. Interestingly, the gene sets that are more susceptible to HO-TRCs (long and highly transcribed) are functionally enriched for multiple oncogenic pathways, and we predict these genes to be hotspots of genomic instability in tissues and tumors experiencing BRCA2 deficiency. Indeed, candidate genes identified by Ok-seq in this study (those genes with new dormant origins at TTSs in BRCA2-KD FTEs), and genes enriched for DNA mutations in *mutBRCA2* patient samples are both similarly affected by hallmark gene transcriptome deregulation for several oncogenic pathways, including EMT. Proteomic analysis of a patient with mutant BRCA2 revealed that the normal fallopian epithelium is enriched for expression of proteins related to focal adhesions and maintenance of the extracellular matrix, while the HGSOC proteome from the same patient showed proteins related to these biological functions are significantly downregulated. This suggests a potential step-wise loss-of-function for EMT-related genes in HGSOC precursor cancer progression. A previous study showed that FTE-derived HGSOCs are enriched for mesenchymal states compared to OSE-derived HGSOCs[4], and EMT-high HGSOCs are associated with worse patient survival[88]. We predict that stalled replication forks due to HO-TRCs occurring at gene termini with persistent R-loops could lead to transcription-associated DNA damage in BRCA2-deficient FTEs and that this may drive gene expression alterations through mutations, improper transcriptional termination, splicing defects, and/or DNA breaks. For example, one candidate gene with new origins downstream of its TTS in BRCA2-KD FTEs is *SEPTIN9* (Supplementary Fig. 4a). *SEPTIN9* functions as a putative breast and ovarian cancer tumor suppressor whose gene products (different mRNA isoforms) have been linked to increased cell migration and ECM degradation in breast cancer models[89]. Therefore, we speculate that HO-TRCs at long, highly transcribed genes could be one mechanism of early STIC lesion development. Finally, we propose a transcriptome-driven mechanism of replication stress, dormant origin firing, and HO-TRCs that are all linked in a cell-type-specific manner. Canonical origins are oriented at TSSs of highly expressed genes, and our results show that the transcriptome likely governs the genomic location of dormant origin firing efficiency as well; this is consistent with previous work that active transcription during G1/S-phase likely clears MCM helicase complexes that were previously licensed/loaded randomly within gene bodies and throughout the genome of different cell types. Therefore, when canonical forks originating at TSSs are stressed and fail to progress, dormant origins redistributed at TTSs of genes incite the potential for HO-TRCs and genomic instability, which can be further exacerbated depending on certain oncogenic drivers or disease backgrounds.

Our results reveal one potential mechanism by which BRCA2 prevents genomic instability and early, FTE-derived HGSOC tumorigenesis. Our results, in accordance with other critical work revealing potential mechanisms by which BRCA2 prevents replication stress (fork protection from degradation, fork reversal mechanisms, RAD51-mediated recruitment)[7–11], reveal that mitigation of replication stress is critical to BRCA2 tumor suppression function. However, the mechanisms of how BRCA2 prevents replication stress specifically at the most prevalent, canonical forks originating near TSSs of high-transcription

genes remains unclear. G4-quadruplex structures are prevalent at TSSs and excessive G4 accumulation leads to replication-stress-induced DNA damage[68], and future work using Ok-seq analysis will reveal molecular and genetic perturbations that cause G4-induced replication stress at TSSs and dormant origin firing. Additionally, BRCA2's role in transcription regulation remains elusive. Both BRCA2 and RNase H2 are required for transcription elongation, and it is unknown whether BRCA2-RNase H2 forms a constitutive complex with RNAP2 to promote transcription elongation at specific genes, i.e. long, transcriptionally active genes. How defective transcription elongation contributes to MCM distribution within gene bodies, replication stress, TRCs, genomic instability, alterations in gene expression, and tumorigenesis remains unclear. Long, transcriptionally active genes are prone to HO-TRCs, and this gene set maintains epithelial integrity in fallopian tube epithelial cells. However, there is currently no evidence to support that transcription-induced DNA damage and mutations in these genes compromise epithelial integrity, and it is unknown to what extent the mechanism proposed herein causes a lack of fallopian tube epithelial integrity to promote early HGSOC tumorigenesis. Understanding the role of BRCA2 in preventing oncogenesis via mechanisms linked to mutagenesis at HO-TRC sites, followed by gene-specific dysregulation of fallopian tube epithelial integrity, will be a challenge for future research.

## Methods

### Key resources and reagents
Key resources and reagents used in this study are described in Supplementary Data 1.

### Tissue culture
FT-194 cells are human fallopian tube secretory epithelial cells (FTEs) that were immortalized and cultured as previously described through hTERT and SV40 large- and small- T antigen transduction[50,90]. FTEs were transduced with doxycycline (DOX)-inducible shRNAs against scramble or BRCA2 sequences using the TRIPZ Inducible Lentiviral system (Horizon) according to the manufacturer's instructions. shScramble or shBRCA2 FTEs were cultured at 37 °C under 5% $CO_2$ in DMEM-F12 (Gibco) in 2% Ultroser G serum (Sartorius) and 1% penicillin/streptomycin (Gibco). FTEs were tested for mycoplasma contamination using a Universal Mycoplasma Detection Kit (ATCC).

### Transfections of plasmids, shRNAs, siRNAs, and cell survival analysis
shScramble/shBRCA2 FTEs were treated with doxycycline to a final concentration of 2 μg/mL every 24 h for BRCA2-KD. Empty vector (EV) (Invitrogen V79020) or human RNase H1 (RH1) (Addgene #111906) plasmids were transiently overexpressed in shBRCA2 FTEs using Fugene HD (Promega) according to the manufacturer's instructions. Cells were harvested for downstream analysis 24 h after EV/RH1 overexpression. For knockdown experiments using siRNAs against control sequences, BRCA2, and RNaseH2A (see Oligonucleotides in Supplementary Table 1), FTEs were treated with Lipofectamine RNAiMax (Invitrogen) according to the manufacturer's instructions. Cells were harvested three days after siRNA transfection for downstream analysis.

### Western blot analysis
Harvested cell pellets were lysed (420 mM NaCl, 0.5 mM EDTA, 20 mM Tris pH = 8, 0.5% v/v Nonidet P-40) on ice for 15 min and lysates were cleared by centrifugation at $14,000 \times g$ for 15 min at 4 °C. The supernatant was quantified by Bradford reagent (Bio-Rad Protein Assay Dye Reagent) according to the manufacturer's instructions. For detection of phosphorylated proteins and their unphosphorylated counterparts, cell pellets were lysed in SDS lysis buffer as previously described[65] and quantified by DC Protein Assay (Bio-Rad). Lysates were loaded in SDS/B-merceptoethanol/glycerol loading buffer onto a NuPAGE 4–12% Bis-

Tris gel (Thermo Fisher) and electrophoresed at 120 V. For detection of BRCA2, gels were transferred overnight at 50 mA at 4 °C to an Immobilon PVDF membrane (Millipore). Otherwise, gels were transferred at 80 V for 2.5 h at 4 °C. Blots were blocked in 5% w/v milk in TBST buffer for 1 h and probed with primary antibody (see Antibodies in Supplementary Table 1) overnight at 4 °C. Membrane was probed for secondary antibody and detected with ECL Prime reagent (GE Healthcare). For the detection of phosphorylated proteins, blots were blocked in 5% BSA in TBST, and both primary and secondary antibody dilutions were made in 5% BSA in TBST.

## Okazaki-fragment sequencing (Ok-seq)

Ok-seq experiments were performed as previously described[27,51]. Briefly, roughly 100-200 million FTEs were pulse-labeled with 25 μM EdU for 2 min and harvested from 15-cm plates at 50–70% confluency. Cell pellets were harvested and lysed (10 mM Tris pH = 8, 25 mM EDTA, 100 mM NaCl, 0.5% v/v SDS, 0.1 mg/mL proteinase K) overnight at 50 °C. DNA was extracted by two sequential phenol-chloroform-isoamyl alcohol extractions, precipitated with ammonium acetate and ethanol, washed with 70% ethanol, and resuspended in TE overnight at 4 °C. DNA was separated by a 5–30% sucrose gradient (generated on Beckman Coulter Optima XE-100) centrifuged for 17 h at 122,000 × g at 20 °C, following 10 min of DNA denaturation at 95 °C and 10 min of incubation on ice. Fractions were collected and a small portion run on a 1.5% alkaline agarose gel. DNA < 200 bp was concentrated by column centrifugation (Ambicon), followed by biotinylation by Click-It reaction (10 mM Tris-HCl pH8, 2 mM CuSO$_4$, 10 mM sodium ascorbate, 2 mM Biotin TEG Azide (Berry Associate)). RNA hydrolysis was performed on DNA (250 mM NaOH at 37 °C for 30 min) and DNA was phosphorylated with T4 polynucleotide kinase (NEB). DNA was prepped for sequencing by two sequential adapter ligations interrupted by pulldown of biotinylated DNA with MyOne T1 Streptavidin Dynabeads (Thermo Fisher) in 10 mM Tris pH 7.5, 1 mM EDTA, 2 M NaCl, and 0.1% v/v Tween-20. Ok-seq libraries were PCR amplified and primer–dimers were removed with SPRI beads (Beckman). Libraries were analyzed by Tapestation (Agilent) and sequenced (paired-end 50 bp) using the NovaSeq 6000 Illumina sequencing platform and SP100 flow cells.

## RNA-seq

RNA was isolated from harvested cells using the RNeasy Kit (Qiagen), followed by DNA digestion with Turbo DNAse (Thermo Fisher) and RNA clean up (RNeasy Kit, Qiagen), all according to manufacturer's guidelines. RNA-seq libraries were prepped using TruSeq RNA Library Prep (Illumina) at the Genome Technology Center at New York University Grossman School of Medicine according to manufacturer's guidelines. RNA-seq reads were trimmed (*Trimmomatic*), aligned to hg19 (*STAR*), and generated to sample count matrix (*featureCounts*) using RNA-star route from the Seq-N-Slide analysis pipeline at New York University Langone Health[91].

## Ok-seq origin analysis

Ok-seq reads were aligned to *hg19* with *bowtie2* and stranded Watson and Crick read counts were binned into 1-kb bins as previously described[27,51] (relevant scripts: see https://github.com/FenyoLab/Ok-Seq_Processing). TSS/TTS genomic annotations for hg19 were obtained from the USCS genome browser (http://genome.ucsc.edu/cgi-bin/hgTables). For Ok-seq origins analysis, the discrete derivative (DER) of smoothed Watson and Crick-strand sequencing data was used to generate an origin score from binned Ok-seq reads. First, 1 kb-binned Watson and Crick Ok-seq reads were Hann smoothed: the W/C binned read count signals were convolved with a Hanning window with a width of 60 kb. This smoothing width was chosen as it provided the most consistency between replicates. Continuous regions where DER(Watson) < 0 and DER(Crick) > 0 were identified

as replication origins and a score was calculated as *DER-SCORE = (−1) × DER(Watson) × DER(Crick)*. The distribution of DER-SCORE was negatively skewed and the least efficient origins were removed after normalization of log$_{10}$(DER-SCORE) by filtering at the lower bound of the full-width-half-maximum of the resulting distribution. These scripts can be found at https://github.com/FenyoLab/okazaki_origins. *Bedtools reldist* was used for relative distance analysis to nearest genomic annotation (TSS/TTS) filtered by gene expression (FPKM) x gene length = gene volume. Statistical analysis comparing the relative distance distribution between conditions uses the output of *bedtools reldist* containing origin abundance at each relative distance and calculates one- and two-sided *t*-tests comparing conditions at various bin sizes across relative distances 0-0.5. For all relative distance calculations presented herein, a bin size = 3 of the first 3 relative distances (0, 0.01, 0.02) was used to compare conditions. This script can be found at https://github.com/FenyoLab/okazaki_origins. *Homer annotatePeaks.pl* was used for genome-wide origins annotation. *Bedtools closest* was used to identify candidate genes with new origins at TTSs in shBRCA2-KD FTEs not present in the corresponding untreated FTEs replicate. Random genomic sites on each chromosome were used as a control and generated to match the number of protein-coding genes per chromosome. All scripts for Ok-seq origin analysis were implemented in Python.

## Proximity-ligation assay (PLA)

Cells seeded on coverslips were fixed in 3.7% v/v formaldehyde for 15 min at room temperature followed by permeabilization with 0.5% v/v Triton X-100 for 5 min at room temperature. Duolink Proximity Ligation Assay (Millipore Sigma) was performed using various antibodies against targets of interest (see Antibody Table), Duolink In Situ PLA probes (Millipore Sigma), and Duolink In Situ Detection Reagents (Millipore Sigma) according to the manufacturer's instructions. Imaging was performed on the Keyence BZ-X710 microscope and the number of foci per nucleus was scored using ImageJ.

## Multi-color single-molecule super-resolution microscopy (SMLM)-STORM

SMLM was performed as previously described[67]. Cells seeded on coverslips were labeled with 10 μM EdU for 10 min in media at 37 °C, fixed in o4% v/v paraformaldehyde for 10 min at room temperature, and extracted with CSK buffer (10 mM HEPES pH = 7.4, 300 mM sucrose, 100 mM NaCl, 3 mM MgCl$_2$, 0.5% v/v Triton X-100) for 3 min at room temperature. Cells were gently washed in PBS three times and blocked in blocking buffer (2% w/v BSA, 2% w/v glycine, 0.2% w/v gelatin, 50 mM NH$_4$Cl) for 15 min at room temperature or overnight at 4 °C. EdU was detected by Click-It chemistry (Thermo Fisher), and coverslips were stained with antibodies against targets of interest (see Antibodies). Coverslips were mounted on imaging slides and washed with imaging buffer (1 mg/mL glucose oxidase, 0.02 mg/mL catalase, 10% glucose, 100 mM cysteamine) prior to imaging.

SMLM Imaging was performed on a custom-built inverted microscope based on the ASI RAMM platform: a 639 nm laser (ultralasers, MRL-FN-639-800) and a 561 nm laser (Coherent, Sapphire 561 LPX-500) were aligned, expanded, collimated, and directed into a TIRF objective (Olympus, UApo N, 100× NA1.49) using a penta-edged dichroic beam splitter (Semrock, FF408/504/581/667/762-Di01). Lasers were adjusted into Highly Inclined and Laminated Optical sheet (HILO) mode prior to imaging with the illumination intensity at ~1.5 kW/cm$^2$ for the 639 nm laser and 1.0 kW/cm$^2$ for the 561 nm laser at the exit of the objective. A 405 nm laser (ultralasers, MDL-III-405-100) was used to drive AF647 to its ground state. AF647 was illuminated by the 639 nm laser, and then JF549 was illuminated by the 561 laser. Both of their fluorescence was expanded to 1.67× and filtered by a single-band pass filter (Semrock FF01-676/37 for AF647, and FF01-607/36 for

JF549). SMLM imaging data was collected using a sCMOS camera (Teledyne Photometrics, Prime 95B) at 33 Hz/30 ms per frame for a minimum of 2000 frames for each color.

A regular DAOSTORM routine was used for single-molecule localization, as previously described[62]. Each frame was stacked with a smaller (σ ~ = 143 nm) and a bigger (σ ~ = 286 nm) Gaussian kernel, and the local maximum was identified by subtracting the bigger from the smaller filtered images. A 9 × 9 square (1 pixel is about 65 nm) around each local maximum was cropped and submitted for MFA sub-pixel localization by fitting the data to one or more 2D-Gaussian point spread functions through maximum likelihood estimations (MLE). The fitting accuracy of each target of interest that was given by the Cramér-Rao Lower Bound was analyzed by fitting its distribution with a skew-Gaussian distribution, using the center as the average localization precision. Localizations that appeared within 2.5 times of the average localization in consecutive frames were averaged and considered as a localization from one blinking event. The coordinates of such events were then submitted for pair-correlation analysis. Representative SMLM images were generated from the coordinate list to the 10 nm pixel canvas and blurred with a Gaussian kernel (σ ~ = 143 nm) for display.

SMLM data analysis, including the alignment of three-color channels, Auto-Pair Correlation (PC), and Cross-PC analysis, was performed as previously described[54,62,65–67]. For more information on relevant scripts, see https://github.com/yiny02/direct-Triple-Correlation-Algorithm.

### DNA fiber analysis

DNA fiber analysis was performed as previously described[67]. Briefly, cells at 50–70% confluency were sequentially labeled for 20 min each with IdU then CldU nucleotide analogs (see Antibodies) at 37 °C, with two quick PBS washes in between. Cells were harvested by trypsinization, washed with cold PBS, and lysed (0.5% v/v SDS, 200 mM Tris pH = 7.4, 50 mM EDTA) for 6 min. DNA lysate was spread on glass slides, tilted at a 15° angle, dried at room temperature, and fixed in cold 3:1 methanol:acetic acid. Prior to staining, DNA was denatured in 2.5 M HCl for 30 min at room temperature, washed with PBS, and blocked in 5% w/v BSA + 0.5% v/v Triton X-100 in PBS. Slides were stained with primary antibodies (mouse anti-IdU [B44] (BD Biosciences 347580), rat anti-CldU [BU1/75 ICR1] (Abcam ab6326)) (see Antibodies) diluted in blocking buffer, followed by PBS washes and fluorescent secondary antibody staining (goat anti-mouse IgG H+L AF488 (Thermo Fisher) and goat anti-rat IgG H+L AF594 (Thermo Fisher)) (see Antibodies). Fibers were visualized using Keyence BZ-X710 microscope and fiber track lengths were calculated using ImageJ by converting a known μm length to pixels, and 1 μm = 2.59 kb[65].

### DNA:RNA hybrid immunoprecipitation (DRIP)-qPCR

DRIP-qPCR was performed as previously described[92] with modification. Briefly, cell pellets corresponding to 5–10 million cells were harvested and resuspended homogenously in TE buffer, followed by addition of SDS to a final concentration of 0.5% v/v and proteinase K to a final concentration of 0.5 mg/mL and incubation at 37 °C overnight. DNA was extracted by two sequential phenol-chloroform-isoamyl alcohol extractions, precipitated in 2.5 M ammonium acetate and ethanol, washed with 70% v/v ethanol, and resuspended overnight at 4 °C in nuclease-free TE. DNA was sheared by sonication (8 cycles of 30 s on/30 s off) in a Bioruptor Pico (Diagenode) to a range of DNA fragment size of 200–1000 bp according to manufacturer's instructions, and sonication efficiency was verified on a 1% agarose gel. 30 μg of DNA was immunoprecipitated overnight by rotation at 4 °C with 3 μg of S9.6 antibody (Kerafast) in binding buffer (10 mM sodium phosphate pH = 7, 0.14 M NaCl, and 0.05% v/v Triton X-100). A slurry of 10 μL of Protein G Dynabeads (Thermo Fisher) + 10 μL of Protein A Dynabeads (Thermo Fisher) prewashed in binding buffer were added to each IP and incubated at 4 °C for two

hours. IPs were washed twice in binding buffer for 15 min each at 4 °C with gentle rotation. DNA was eluted in 50 mM Tris pH = 8, 10 mM ETDA pH = 8, and 0.5%v/v SDS for 1 h at 55 °C with agitation. Eluant was recovered and DNA was purified using phenol-chloroform-isoamyl alcohol in 2-mL 5PRIME phase-lock gel tubes (Quantabio) according to manufacturer's instructions, followed by DNA precipitation and resuspension in RNase-free TE buffer. qPCR was performed on diluted IP'd DNA at the loci of interest (see Oligonucleotides) as previously described[93] on a QuantStudio 6 Pro qPCR machine (Applied Biosystems). Data was analyzed as a percent of input fold change over control condition[92].

### Data access and gene set enrichment analysis

Ovarian Serous Cystadenocarcinoma samples from the Firehose Legacy of The Cancer Genome Atlas (TCGA) were accessed by cBioPortal (https://www.cbioportal.org/) on July 19, 2023. mRNA expression data (RNA-Seq V2 RSEM) and mutation data from six samples with *BRCA2* mutations were compared to 158 samples with *TP53* mutations to identify genes with mutations upregulated in mut*BRCA2* samples. Genes with somatic sequence alterations in STIC lesions and HGSOCs were accessed from all nine patients from Labidi-Galy et al.[2]. Proteomic data from the Clinical Proteomic Tumor Analysis Consortium (CPTAC) from normal and HGSOC tissue of one *mut*BRCA2 patient were accessed from McDermott et al.[94]. Molecular signature database (MSigDB, https://www.gsea-msigdb.org/gsea/msigdb/index.jsp) was used to determine hallmark gene set and KEGG pathway annotation from the gene set enrichment analysis database (GSEA, UC San Diego and Broad Institute).

### Statistics and reproducibility

Primary data were recorded using Microsoft Excel and statistical analyses were performed using GraphPad Prism 8, FlowJo 10, ImageJ 1.52a, Matlab (v2017b), and OriginLab (2018) software. Exact p-values are provided for each experiment in the Source Data file associated with this article. A biological replicate dataset was obtained for all Ok-seq samples. All SR experiments were performed at least in duplicate with $n > 60$ sample size. All IF experiments, including DNA fiber analyses, are based on three biological replicates with a sample size $n > 200$. Western blotting experiments were performed in at least two independent experiments. Source data contains all raw data used for statistical analysis.

### Reporting summary

Further information on research design is available in the Nature Portfolio Reporting Summary linked to this article.

## Data availability

The datasets generated in this study have been deposited in NCBI's Gene Expression Omnibus and are accessible through GEO Series accession number GSE239858. The results in Fig. 5F–G are in whole or part based upon data generated by the TCGA Research Network: https://www.cancer.gov/tcga. Further information, resources, and reagents are available from the corresponding author upon request. Source data are provided as a Source Data file. Source data are provided with this paper.

## Code availability

All Ok-seq analysis code is publicly available under the GNU General Public License v2.0 in our GitHub repository: https://github.com/FenyoLab/Ok-Seq_Processing and https://github.com/FenyoLab/okazaki_origins. Code relevant to SMLM analysis is provided here: https://github.com/yiny02/direct-Triple-Correlation-Algorithm. Documentation is provided as a readme file, and specific instructions on parameters to functions are embedded as inline comments in the code.

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

## Acknowledgements

We thank members of the Huang, Fenyo, and Rothenberg labs for technical assistance and critical discussions. L.G. was supported by the NYU MSTP Scholar Award; R.D. by Dr. Miriam and Sheldon G. Adelson Medical Research Foundation, and E.R. and T.T.H. by the generous gift from Laura Chang and Arnold Chavkin, and the Goldberg Family Foundation. This work was supported by NIH grants: GM139610 and ES031658 (T.T.H.); R35 GM134918 (D.S.); R35 GM134947, AI153040, and CA247773 (E.R.); CA270788 and CA215990 (R.B.J.); and P50 CA228991 (R.D.). Sequencing was performed at the NYU Genome Technology Center (RRID: SCR_017929). Additional research support was provided by a V Foundation BRCA-Convergence Team Award to T.T.H., R.B.J., and E.R. The NYU Genome Technology Center shared resource is partially supported by the Cancer Center Support Grant P30 CA016087 at the Laura and Isaac Perlmutter Cancer Center.

## Author contributions

L.G. and T.T.H. conceived and designed the research project. L.G., S.L., and D.G. performed the SMLM imaging studies. R.D., J.J-S., and R.B.J. generated, developed, designed, and validated BRCA2-deficient FTE cells. L.G., S.K., W.X., and M.K. performed the research and collected data. L.G., S.K., S.L., D.F., D.J.S., E.R., and T.T.H. analyzed and interpreted the data. L.G. wrote the initial draft of the manuscript, and L.G., S.K., E.R., D.F., and T.T.H. edited/revised the manuscript.

## Competing interests

The authors declare no competing interests.
