## [Peer Review File · Nature Communications]

Dormant origin firing promotes “head-on” transcription-replication conflicts at transcription termination sites in response to BRCA2 deficiencyEditorial Note: Parts of this Peer Review File have been redacted as indicated to remove third-party material where no permission to publish could be obtained.

REVIEWER COMMENTS

Reviewer #1 (Remarks to the Author):

In the manuscript entitled "Dormant origin firing promotes "head-on" transcription-replication conflicts at transcription termination sites in response to BRCA2 deficiency", the authors describe a new mechanism by which tumor suppressor protein BRCA2 protects fallopian tube epithelium from transcription-driven genomic instability. Using OK-seq, the authors show that BRCA2 ensures efficient replication fork progression thereby suppressing the activation of dormant replication origins around transcription termination sites of genes. Using PLA and SMLM approaches, they demonstrate that dormant origin firing in BRCA2-deficient cells leads to transcription-replication conflicts (TRCs), R-loop formation and associated genomic instability. Interestingly, the authors find that BRCA2 beyond its known role in fork protection can also recruit RNase H2 to resolve R-loops and prevent TRCs. Overall, the authors present a compelling story that is novel and of interest to the community. Nevertheless, some additional experiments will be needed to strengthen the point the authors want to make, especially about the strong orientational claim that this is specific to HO vs CD conflicts as well as the recruitment of RNase H2.

Major points:

1. In Figure 1B the authors aim to show that HU treated RPE1 cells show increased origin firing at TSS of high-volume genes. This result is not visible in the figure for me. Employing statistical analysis might be generally helpful for these approaches.
2. In Figure 1E also less negative enrichment can be seen in 3'UTR which is of course close to the TTS region. The fold-change is similar to the one at the TTS How do the authors explain this? Are those intragenic dormant origins not fully pushed off the gene by transcription? The authors claim that there was no effect on intragenic origins. How does this fit?
3. For all PLA data: Statistics on PLA data should be performed on the mean foci values of the biological replicates, not the pooled data of all biological replicates. The observed 200-300 cells are not independent observations (since they come from only 3 slides/wells) which violates a central assumption of the Mann Whitney test performed.
4. For PLA assays showing increased PLA foci upon BRCA2 KD – Please show S-phase specific increases in PLA with a combined EdU-Click plus PLA staining. At least for the most relevant Ser2P-PCNA combination.
5. The authors use Ser2P and Ser5P as an approach to distinguish between CD (Ser5P) and HO (Ser2P) conflicts. As this represents a new approach to distinguish between the orientation of TRCs, some further validation would be important for correct interpretation of these results. The authors could try to repeat the PLA assays with an independent Pol II CTD modification antibody that shows a similar enrichment at TTS as Ser2P. For example Pol II Thr4P is a mark of transcription termination and increased PLA foci of Pol II Thr4P and PCNA would be a complementary approach and thus strengthen the idea that more TTS-linked marks such as Ser2P and Thr4P are reliable markers of HO conflicts.
6. Figure 2E/F: The authors use SMLM as a powerful and complementary approach to PLA. The data is visualized as Pol II S2P close to EdU. It is unclear to me as a non-expert on SMLM what "near" is. To how many nanometers distance does this correspond and how has this threshold for being near been set? Can one come up with a combined proxy for sites near Pol II and EdU positive regions? It would also be important to show more of the raw data, for example the total level of EdU incorporation, potential differences in S-phase stages of the cells, etc as such parameters may also bias the results.
7. The authors have performed DRIP-qPCR analysis of well-characterized R-loop genes (Fig S4D). However, only one gene is shown (ACTB). 2-3 additional examples are needed to support the idea that BRCA2 KD increases R-loop levels at the TTS. Moreover, it remains unclear to me if ACTB is indeed one of the genes which shows the dormant origin firing phenotype at the TTS. The authors could take advantage of their OK-Seq dataset and confirm the phenotype on a few additional example "dormant origin firing" genes.
8. BRCA2 KD in combination with additional RNaseH1 OE to rescue R-loop levels would be needed to strengthen the point of BRCA2 deficiency induced R-loops at the TTS. This would also further support

the point made in the PLA in Figure 4A.

9. Rescue experiments are first executed by RNaseH1 overexpression, followed by RNaseH2 depletion experiments in another cell line (U-2OS). There is no rationale provided by the authors for this rather abrupt transition in the manuscript. Can the authors reproduce their findings in U-2OS cells with RNaseH2 in FTE cells?

10. Figure 4D – again as a non-expert in this technique: How close does RNH2 need to be to be considered “near”? How do the results change upon changing that distance parameter?

11. The authors propose that BRCA2 and RNaseH2 functionally cooperate to mitigate HO TRCs. This is a very interesting idea and supported by the synthetic lethality. Do both proteins actually interact? Can this be validated by e.g Co-IP Western blot analysis? Related to this, the authors can also check for a potential interaction of BRCA2 with RNaseH1.

12. It would be important to show that global DNA synthesis or total levels of Ser2P/Ser5P are not affected by the used concentration of CDC7i. Moreover, I disagree with the assumption that CDC7i will exclusively inhibit dormant origins, rather new origin firing in general. This should be revised in the manuscript.

13. ATRi inhibition also increases the TRC PLA foci of Ser5P-PCNA (Fig S4F). It is difficult for me to conclude from this that ATRi necessarily produces CD TRCs due to more firing at canonical origins. ATRi-driven HO TRCs seen with Ser5P-PCNA could also stem from cryptic intragenic origins and HO TRCs. Since it remains technically impossible to determine from what genomic location a PLA focus originated from, I would suggest that the concept that Ser2P equals HO TRC and Ser5P equals CD TRCs should be more carefully discussed throughout the manuscript.

14. The proximity analysis between Pol II Ser2P and various DNA damage markers with and without CDC7i is a strong point of the story. It would be nice to validate a few of those with SMLM to ensure the damage localizes to the TRC region directly.

Minor points:

- In the abstract the authors mention that “BRCA2 is a tumor suppressor protein responsible for safeguarding the cellular genome from replication stress and genotoxicity, but the mechanism(s) by which this is achieved remains elusive.” This should be rephrased as there is a significant amount of literature on the multiple mechanisms how BRCA2 safeguards the genome, e.g. its function to bind to RAD51 to promote HR, as also mentioned by the authors in the limitations of the study.
- In the introduction (line 8) the authors state that endogenous sources of DNA damage drive genomic instability and cancer progression in FTEs despite them being unclear. Is this known from literature before this study? Please cite or rephrase.
- Figure 1A, it would help the readability of the figure to provide an X-axis labels on the TSS to TTS relative distance plot to make it more easy to grasp what is shown here.
- Figures S1G/H would profit from figures titles marking them as high/low volume genes
- On page 7 in line 33 the authors state that Ser5P phosphorylation of the CTD occurs to promote imitation and promoter proximal pausing a few kilobases downstream of the TSS. This statement needs to be revised. First, promoter proximal pausing occurs 50-70 bp downstream of the TSS not kilobases. Moreover, the Ser5P mark is to my knowledge highly present at paused Pol II but not directly implied in promoting pausing in comparison to elongation factors like DSIF or NELF.
- On page 8 line 2 the authors say that Ser2P Pol II accumulates downstream of gene bodies. This should also be rephrased as Pol II Ser2P starts accumulating downstream of the pausing site throughout the gene body with highest levels before the TTS. In my view, downstream of the gene body would be outside of the gene already.
- The authors state “Considering intragenic origin firing is suppressed in actively transcribed genes (Figures S2B-F), the proximity between various phospho-forms of RNAP2 and active forks arising from intergenic origins marks distinct TSS- and TTS-oriented locations for TRC events”. This statement should be rephrased as an hypothesis and not considered a fact since no existing literature has convincingly proven this assumption yet.
- For the PLA experiments in Figure 3C statistics should be provided like in all other PLA experiments.
- Figure 5B and 5C - Quality of track images/contrast could be improved.
- In the discussion the authors speculate that reduced Ser5P-PCNA interaction in PLA could be

stemming from decreased canonical origin usage upon BRCA2 deficiency. With their OK-seq data set at hand the authors are uniquely positioned to address this question. This analysis would provide crucial insights into a shift from CD to HO conflicts at highly transcribed genes upon BRCA2 loss.

Reviewer #2 (Remarks to the Author):

Goehring et al use OK-seq, PLA and superresolution microscopy to study the effects of BRCA2 depletion on origin firing and transcription-replication collisions (TRCs) in fallopian tube epithelium (FTE) cells, the likely cell-of-origin for high-grade serous ovarian carcinoma (HGSOC). They propose that forks originating from TSSs of transcriptionally high-volume genes are more inclined to slow-down or stall in BRCA2-deficient cells, resulting in increased downstream origin firing and increased head-on (HO) TRCs in the 3' end of these genes. They propose that BRCA2 can directly mitigate R-loops at HO-TRCs through the recruitment of RNaseH2, and indirectly suppress HO-TRCs through the passive inhibition of replication stress-induced dormant origin firing. They find that that endogenous sites of HO-TRCs are enriched for gamma-H2AX, phosphorylated RPA, and R-loop accumulation, which can be mitigated by limiting dormant origin firing with inhibitors of the origin firing kinase Cdc7. Finally, a bioinformatic analysis suggests that the affected genes are enriched in several oncogenic pathways including the epithelial-mesenchyma transition, suggesting possible pathways for HGSOC progression. Overall, this is an impressive study rich in novel data. However, I found that several experiments were insufficiently explained. Furthermore, the proposal that BRCA2-depletion results in head-on rather than codirectional collisions is in my opinion insufficiently demonstrated.

1. The description of the "new computational methodology to analyze our previous Ok-seq dataset to probe for replication origins from Ok-seq reads mapped to the Watson and Crick strands" is complex and not sufficiently clear and illustrated (page 5, lines 17-21, Fig 1 A, Fig S1A). The authors compute increments in Crick or Watson Okazaki fragment abundance between consecutive 1-kb windows, compute the log10 of the inverse product of these two measurements, smooth the results using a Hann function with a 60 kb window and filter out the origins whose score is less than the lower bound of the full width at half maximum (FWHM) of the score distribution. This was tentatively illustrated in Figure 1A and S1A but in a too simplistic manner (e.g. the Hann smoothing is not explained). First, the authors should illustrate each of these steps for a large chromosomal segment (say, one or a few Mb) containing large transcribed genes whose origin distribution is affected by hydroxyurea treatment. They should plot i) the Watson and Crick Okazaki fragment abundance profiles in the absence or presence of hydroxyurea; ii) the replication fork directionality ($RFD = (C-W) / (C+W)$) as in previous publications; iii) the unsmoothed discrete derivatives by 1 kb windows $DER(Watson)$ and $DER(Crick)$; iv) the Hann-smoothed profile(s); and v) the position of origins retained after filtering. Thus, the readers will be able to judge by themselves, on this exemplary locus, the magnitude of the effects of hydroxyurea, from raw to processed data, and to connect the origin scores with the actual changes in Okazaki fragment abundance and RFD profiles. Second, I anticipate that changes in Okazaki fragment abundance in successive 1-kb windows predominantly reflect statistical noise in read counts rather than actual changes in fork direction. The derivative of a noisy signal being unstable and not meaningful, the Hann smoothing step is crucial but may affect the precision of mapping origin activity changes. In short, do the observed score changes reflect global changes over a 60 kb window or can they be confidently attributed to narrower (1-kb ?) windows ?

2. The "relative distance calculation" procedure used to decide whether origins map at TSSs or near TSSs or are "not oriented around TSSs" was not sufficiently explained (page 5, line 25). Why use a relative rather than an absolute distance measurement ? Relative to what ? To gene length ? How is it computed ? Why not simply provide the absolute distance between origins and the nearest active (or inactive) TSS ? In Fig 1B,C,F,G, and many similar Figures throughout the paper, is it the absolute value of the distance or the arithmetic distance that is plotted from 0 to 0.4? In other words, do these

Figures only show the distribution of origins inside the gene, or a superimposition of origins inside and outside the gene ? In the former case why only show intragenic origins ? In the latter case it would be preferable to separately plot extragenic and intragenic origins using positive and negative relative distances rather than their absolute values. Please clarify.

3. The authors state that there is "a striking increase in origins oriented at TTSs of high-volume genes compared to untreated conditions (Figure 1C)." However, looking at the graduations of the y-axis, this increase is only from 2.2% to 3.2% of origins mapping at the TTS. While this seems significant, it is only a weak effect, somewhat visually amplified because the y-axis does not start at 0. In Figure 1G the increase in origins at TTSs following BRCA2 depletion seems even smaller (from 2.0% to 2.3%). Please replace "striking" by an objective quantitative measurement.

4. I could not find the Extended Supplement Data with exact distances between TTSs and new origins in BRCA2-KD that is said to support the identification of candidate genes with new origins near TTSs (page 6, line 27).

5. Figure 1I, right panel, shows the appearance of new intragenic origins following BRCA2-KD, in apparent contrast with the statement that "BRCA2-KD did not impact levels of intragenic origin usage (Figure 1E)" (page 7, lines 32-34). Please explain.

6. Page 8, line 11, reads "Interestingly, pSer5-RNAP2 proximity to PCNA decreased in BRCA2-KD (Figure 2C), consistent with a previous study that found BRCA2-deficient cells have decreased pSer5-RNAP2 signal at promoters of transcriptionally-active genes (ref 23)". However, Reference 23 shows that BRCA2 inactivation by depletion or cancer-causing mutations triggers RNAPII accumulation at promoter-proximal pausing (PPP) sites in actively transcribed genes (Fig 3A in Ref 23). It is not clear if this accumulated RNAPII is phosphorylated on Ser 5, Ser 2, or both, and the small decrease in the Ser5 form seen on their Figure 3B at PPP does not appear statistically significant, in contrast to total RNAPII accumulation. Please clarify, or remove the part of the sentence "consistent with a previous study..."

7. Figure 2E,F, please clarify whether what is shown is a nuclear slice (in this case, which thickness ?) or the projection of the total nuclear signal on a plane. This is important to evaluate the significance of overlapping signals. Please also clarify what is the unit on the y-axis. I wondered whether the total RNAP2 signal associated with EdU tracks would increase, decrease or remain constant following depletion of BRCA2. Can the authors perform their assay with their pan-RNAP2 antibody ? Finally, the length of DNA labelled during a 10 min EdU pulse should be about 10-20 kb, but shorter if replication was stalled for a few minutes. I would expect RNAP2 to be localized at the border of (on average, shorter) EdU tracks in the case of stalling collision, but anywhere along (full-length) EdU tracks in the case of a moving RNAP2 transcribing a newly replicated DNA stretch without stalling. Does the resolution of the technique allow the authors to discriminate these two scenarios? In other words, do coincident EdU and RNAP2 signals overlap differently for pSer5 and pSer2 forms of RNAP2, and are EdU signals that overlap with RNAP2 less intense than EdU signals that do not overlap with RNAP2?

8. Page 9, lines 1-4. The interpretation that forks generated from TTS origins converge with elongating RNAP2 to increase HO-TRCs is likely but not proven. PLA and SMLM experiments suggest that BRCA2 depletion increases colocalisation of RNAP2 with PCNA and with newly synthesized DNA, but do not discriminate between head-on and co-directional collisions. One could for example argue that a significant fraction, or even a majority, of these colocalisation events correspond to co-directional collision of forks emitted from the TSS, and that a consequent fork slowing would give more time for origins at and downstream of the TTS to fire and rescue the stalled co-directional forks. Can the authors formally exclude this (or other alternative hypotheses) and give more convincing evidence for head-on collisions ? Same criticism for the interpretation of the effect of hydroxyurea, Fig S4A, page 9, lines 10-12.

9. Page 9, lines 20-22, " the PLA signal between BRCA2 and PCNA were increased upon HU treatment, revealing the elevated recruitment of BRCA2 to active forks" : did they mean stalled forks ?

10. DNA fiber analyses in Fig 5. It has been shown that measurements of IODs and fork speeds can be strongly biased by the length of DNA fibers analyzed (Técher et al PMID: 23557832). It is therefore important to show that fiber size distribution is similar in different samples to allow meaningful comparisons. This important control was apparently not performed.

Reviewer #3 (Remarks to the Author):

In this manuscript, the authors interrogate whether and how replication stress promote genome instability and cancer progression. They do so in the context of BRCA2 deficiency, in a relevant model of high-grade serous ovarian carcinoma (HGSOC), of which a good fraction is homologous recombination deficient (HRD).

Given the role of BRCA2 in preventing or suppressing R-loops previously established, and in particular the role of BRCA2 in recruiting RNaseH2 to DNA-RNA hybrids at sites of DSBs, and previous report on the regulation of transcription by BRCA2, they investigate whether BRCA2 deficiency affects replication fork dynamics and in particular, transcription-replication conflicts that could explain the genome instability that fuels tumorigenesis in this setting.

They found that BRCA2 KD leads to firing of dormant origins specifically at TTS of highly expressed long genes, "high-volume genes". Then they tested whether the increased of origin firing lead to the presence of HO-TRCs using the proximity of initiation or elongating RNA pol II and PCNA as a proxy which can they show can be alleviated by PARPi or CDC7i. Finally, they show that the recruitment of RNaseH2 to HO-TRCs to resolve R-loops at HO-TRCs is dependent on BRCA2 implicating the latter in this function.

Overall this an interesting manuscript showing that dormant origin firing induce replication stress in BRCA2 deficient cells ultimately resulting in HO-TRCs at long highly transcribed genes although the relation to RNaseH2 is unclear. The authors further suggest this is particularly relevant in HGSOC oncogenesis for which they provide certain evidence based on patient data.

I think this manuscript has potential but would require major revision for publication, see specific comments below:

Major:

1. The fact that elongating RNAPol II colocalize with EdU or PCNA is indicative of head-on replication-transcription collisions seems far-fetched. A minimum control in these assays (Fig. 2) would be to test whether RNaseH treatment reduces this colocalization in PLA and SMLM experiments the way they did then in Fig. 4A. Moreover, if the idea is that RNaseH2 deals with these conflicts, RNaseH2 overexpression should be employed to control for this and confirm this is the case.

2. Although both PLA and SMLM seem to go in the same direction, they are based on the same principle. An orthogonal method such as the use of DRIP (DNA-RNA hybrid immunoprecipitation) using primers at TSS vs TTS would be more convincing also to show that these transcription conflicts occur close to TTS and at loci where the authors have mapped the new active origins (Fig. 1I). This experiment has been performed and is somewhat hidden in Fig. S4D however only with a single locus, minimally this should be done in 2-3 to show that is truly general for the genes affected and it does not happen at "low volume genes". Importantly, is the RNaseH1 treatment I Fig. S4D done in the 3d sample or in the 0d sample? It is not clear from the legend nor the figure and it is critical as they need to show that the one occurring at 3d (BRCA2 depletion) is sensitive to RH1 treatment.

Also, validating the proxy with the Serp5 and Serp2 using the proximity between Serp2 and Serp5 with DNA-RNA hybrids would be more convincing.

These experiments are important because all the rest of the manuscript is based on the assumption

that what they observe by PLA are HO-TRC.

3. For all PLA experiments, the signal images and quantification of the antibody alone controls should be shown in the same graph, this is important to discard the signal of the the probes for the two antibodies together is not the sum of the background signal coming from both single antibody controls.

4. The paragraph explaining Fig. 3C. does not match the results shown in the image and in the graph. First in lane 21, authors state PLA signal for BRCA2-PCNA is increased upon HU (fork stalling) which it has indeed been reported before. However, in lane 25, they state the opposite: "fork-stalling alone is insufficient to promote BRCA2 recruitment at the forks". All of this for a graph where there is no statistical analysis.

In addition to showing the statistics this part needs clarification.

5. Fig. 4B, here the authors state: "depletion of RNaseH2A alone, was comparable to the co-depletion of RNaseH2A with BRCA2, suggesting that RNaseH2A and BRCA2 likely work in the same pathway to suppress HO-TRCs at TTS".

Here the authors do not show the relevant statistic comparison, i.e., between siBRCA2 vs siRNH2A alone or siRNH2A vs siRNH2A+siBRCA2 so this is a far reached conclusion. In addition, we do not know that these are HO-TRCs and where they take place (see comment 2).

6. In Fig. 4D and 4E, the authors show that RNaseH2A colocalizes less with elongating pol II in BRCA2 depleted cells and its localization is increased after HU treatment. The authors conclude that both BRCA2 and RNaseH2A cooperate to mitigate toxic HO-TRC. Again, a far-fetched conclusion since they do not show any evidence that BRCA2 and RNaseH2A are in proximity in any conditions. In Fig. 5 they show ATRi exacerbates HO-TRC independent of BRCA2 status so it would be a good condition to test PLA probing RNaseH2A and BRCA2.

Moreover, describing these figures authors state that "both BRCA2 and RNaseH2A in FTE cells were more cytotoxic compared to FTEs treated with either siControl, siBRCA2, or siRNaseH2A alone". So, in Fig. 4B depleting RNaseH2A leads to the same outcome as siBRCA2+ siRNaseH2A leading the authors to state they likely function in the same pathway, and now in Fig. 4F, they show the opposite, namely, synthetic lethality, i.e., different pathways, as previously reported; which one is the right one? How do authors explain these discrepancies?

7. Interestingly, the authors show that CDC7i, which inhibits dormant origin firing, reduce the proximity of elongating Pol II and PCNA only in BRCA2 depleted cells (Fig. 5D). This is an important experiment to relate dormant origin firing observed in BRCA2 deficiency and the possible link with HO transcription-replication conflicts. Again, it should be performed in RNaseH1 (or RH2) treatment conditions to confirm the presence of RNA-DNA hybrids.

8. The authors state: "CDC7i treatment decreased origin density in both control and BRCA2-deficient FTEs: We showed that decreased origin density caused by CDC7i in BRCA2-deficient cells can occur without compensatory increase in fork speed (Figures 5B and 5C). This suggests that a major role for BRCA2 in protecting stalled forks is independent of changes to dormant origin firing".

This interpretation seems too simplistic or wrong. Given the different functions of BRCA2, and the replication stress induced by BRCA2 KD already, I do not see how the authors can infer any function of BRCA2 based on these results.

Nevertheless, the result with CDC7i is important so they should confirm that there is no impact of CDC7i on origin firing at TSS in BRCA2 deficient cells as shown in fig. 1B for HU. This is to discard the possibility that there is a global effect of the inhibitor.

9. The conclusion that R-loops persist at HO-TRCs due to the inability of BRCA2 to recruit RNaseH2 to these sites is also far-fetched (lane 4 before Discussion), see point 6.

Minor:

-The error bars in the BRCA2 KD cells in Fig. 2E are quite large so the results here are not convincing. Doing the same experiment with PCNA instead of EdU as in the PLA setting might be a way to reduce the variability.

- It is not clear to me why they choose to compare BRCA2-mutated to p53-mutated HGSOC cases from TCGA.

- Only a single BRCA2 patient (out of 2) shows EMT pathway enriched

- Pages do not have numbers

Here is our point-by-point response to the Reviewers' Comments (Author's comments/rebuttal in blue):

Reviewer #1 (Remarks to the Author):

In the manuscript entitled "Dormant origin firing promotes "head-on" transcription-replication conflicts at transcription termination sites in response to BRCA2 deficiency", the authors describe a new mechanism by which tumor suppressor protein BRCA2 protects fallopian tube epithelium from transcription-driven genomic instability. Using OK-seq, the authors show that BRCA2 ensures efficient replication fork progression thereby suppressing the activation of dormant replication origins around transcription termination sites of genes. Using PLA and SMLM approaches, they demonstrate that dormant origin firing in BRCA2-deficient cells leads to transcription-replication conflicts (TRCs), R-loop formation and associated genomic instability. Interestingly, the authors find that BRCA2 beyond its known role in fork protection can also recruit RNase H2 to resolve R-loops and prevent TRCs. Overall, the authors present a compelling story that is novel and of interest to the community. Nevertheless, some additional experiments will be needed to strengthen the point the authors want to make, especially about the strong orientational claim that this is specific to HO vs CD conflicts as well as the recruitment of RNase H2.

We appreciate the Reviewer's enthusiasm for our work.

Major points:

1. In Figure 1B the authors aim to show that HU treated RPE1 cells show increased origin firing at TSS of high-volume genes. This result is not visible in the figure for me. Employing statistical analysis might be generally helpful for these approaches.

Shown below is blown up graph of relative distance calculation of origin calls immediately at/near the TSS of Fig 1b. Although this data is not significant by statistical analysis, the trend is consistent with previously published Ok-seq analysis from our group (Chen et al, NSMB 2019). The statistical analysis for the relative distance calculation of origin orientation around genomic annotations is included in each relative distance calculation figure (Fig. 1b-c, Fig.1f-g, new Supplementary Fig. 2, and new Supplementary Fig. 3b-c), and the explanation and scripts used to perform statistical analysis related to these figures has been expanded in the methods (Ok-seq origin analysis) as well as uploaded to the Github repository (https://github.com/FenyoLab/okazaki_origins).

Rebuttal Figure 1 (Corresponding to Fig. 1b) Relative distance origin analysis of previously published untreated and HU-treated RPE-1 cells (Chen et al., 2019)²⁷ relative to TSSs of transcriptionally high-volume genes (FPKM x length > median). N = # of origins per condition. Percent of origins at each relative distance were fit to a nonlinear regression curve (solid line) with 90% confidence interval (shaded). For all relative distance calculations, p values are calculated using paired t-test between conditions for the first three relative distance bins (0, 0.01, 0.02) (see Methods of text).

2. In Figure 1E also less negative enrichment can be seen in 3'UTR which is of course close to the TTS region. The fold-change is similar to the one at the TTS How do the authors explain this? Are those intragenic dormant origins not fully pushed off the gene by transcription? The authors claim that there was no effect on intragenic origins. How does this fit? The Reviewer is correct in assessing that there is less negative 3'UTR origins with BRCA2-KD. This could be due to the fact that we define an origin as 20kb, and the average 3'UTR size is anywhere between 0.5-3kb (PMID: [26597575](https://pubmed.ncbi.nlm.nih.gov/26597575/)). This result could be due to poor resolution in the HOMER software distinguishing between 3'UTR and TTSs (which is defined

as a more discrete location and not a range). Also, while intragenic origin firing in transcriptionally active genes is very, very low, it's not non-existent (**new Supplementary Fig. 3d**), so a portion of intragenic origins at transcriptionally active genes could explain the less negative increase at 3'UTRs. Due to the statistically significant impact at TTSs (**Fig. 1g**), we chose to focus our attention here.

3. For all PLA data: Statistics on PLA data should be performed on the mean foci values of the biological replicates, not the pooled data of all biological replicates. The observed 200-300 cells are not independent observations (since they come from only 3 slides/wells) which violates a central assumption of the Mann Whitney test performed.

Thank you for this important comment. We have updated our statistical analysis of all of our PLA experiments, and have now included the appropriate statistical analyses of the mean of each biological replicate for all PLA experiments in both the main Figures and Supplementary Figures.

4. For PLA assays showing increased PLA foci upon BRCA2 KD – Please show S-phase specific increases in PLA with a combined EdU-Click plus PLA staining. At least for the most relevant Ser2P-PCNA combination.

We appreciate the Reviewer's comment. We have now included new experiments showing pSer2 RNAP2 – PCNA PLA and pThr4 RNAP2 – PCNA PLA with the addition of an 10 minute EdU pulse to mark S-phase cells. The results are consistent with our findings and shows increased TRCs (PLA signal) with BRCA2-loss in replicating cells (EdU-positive). This is now part of **new Supplementary Fig. 5a**.

Rebuttal Figure 2 (Corresponding to new Supplementary Fig. 5a). PLA of pSer2-RNAP2 and PCNA of pulse-labeled EdU+ FTEs +/- shBRCA2. Data presented shows 200 nuclei from 3 biological replicates. p values calculated using unpaired two-tailed t-tests. Left: number of foci per nucleus. Right: average number of foci per nucleus per biological replicates. Error bars = mean, std. Scale bar = 10 μ m in representative pictures.

5. The authors use Ser2P and Ser5P as an approach to distinguish between CD (Ser5P) and HO (Ser2P) conflicts. As this represents a new approach to distinguish between the orientation of TRCs, some further validation would be important for correct interpretation of these results. The authors could try to repeat the PLA assays with an independent Pol II CTD modification antibody that shows a similar enrichment at TTS as Ser2P. For example Pol II Thr4P is a mark of transcription termination and increased PLA foci of Pol II Thr4P and PCNA would be a complementary approach and thus strengthen the idea that more TTS-linked marks such as Ser2P and Thr4P are reliable markers of HO conflicts.

This is an excellent suggestion by the Reviewer. We performed this experiment as suggested by the Reviewer and showed that pThr4 RNAP2 – PCNA PLA signal, similar to Ser2P, also increases with BRCA2-KD in EdU+ cells. This is now part of **new Supplementary Fig. 5b**.

Rebuttal Figure 3 (Corresponding to new Supplementary Fig. 5b). PLA of pThr4-RNAP2 and PCNA of pulse-labeled EdU+ FTEs +/- shBRCA2. Data presented shows 200 nuclei from 3 biological replicates. *p* values calculated using unpaired two-tailed t-tests. Left: number of foci per nucleus. Right: average number of foci per nucleus per biological replicates. Error bars = mean, std. Scale bar = 10 μ m in representative pictures.

6. Figure 2E/F: The authors use SMLM as a powerful and complementary approach to PLA. The data is visualized as Pol II S2P close to EdU. It is unclear to me as a non-expert on SMLM what "near" is. To how many nanometers distance does this correspond and how has this threshold for being near been set? Can one come up with a combined proxy for sites near Pol II and EdU positive regions? It would also be important to show more of the raw data, for example the total level of EdU incorporation, potential differences in S-phase stages of the cells, etc as such parameters may also bias the results.

Upon reading the reviewers' questions we realized that our choice of using the term "near" to describe the metrics derived from the co-localization analysis of our SMLM data, have understandably resulted in much confusion. This has propelled us to modify the manuscript and replace the term "near" with the term "at" which are exchangeable in describing the derived metrics and will prevent further confusion to non-expert readers. Beyond these changes, we also provide the reviewers with a brief note regarding the SMLM analysis that explains the initial reasoning for using the term "near" to describe the co-localization of two-color SMLM data. However, we emphasize that the present work builds on established methods, assays and computational tools that are described in great detail in our previous publications and well as work from other labs (see PMID: 21926998, PMID: 22384026, and particularly PMID: 35365626, see **Methods Correlation functions for analysis of SMLM data, and Supplementary Note 1, PMID: 34473946, Supplemental information, Methods S3-S4, and also PMID: 30631072 SI note 2, and Supp Fig 2. PMID: 25843623**) and all the technical and experimental parameters are now well-established and have been thoroughly evaluated by expert reviewers and meticulously validated in prior works. Since the focus of the current study is the delineation of molecular mechanisms using these and other approaches, rather than the re-validation of the already established SMLM assays and analyses, we limited our descriptions of the methods and instead reference specific sections from our prior studies that provide in-depth and elaborated technical descriptions that fully addresses all the questions raised by the reviewers. We also provide relevant snippets from these publications below.

Brief explanation of analyses two-color SMLM signal co-localization and related terminology (see PMID: 21926998, PMID: 22384026, and particularly PMID: 35365626):

SMLM images present the coordinates of individual fluorescent molecules that were localized with an accuracy of several nanometers (shown as single pixels). SMLM data substantially differs from the data generated via conventional epi-fluorescence or confocal microscopy approaches that measure diffraction-limited intensity distribution over multiple pixels, where measurements of co-localization or overlap between two objects/foci/clusters typically relies on simple segmentation of clusters boundaries. In SMLM data, clusters are made up of individual molecules, which requires different approaches for calculating co-localization. In the manuscript we utilize a robust and unbiased data-mining statistical approach that measures the pair-wise distances between every molecule of one color to all the molecules in a different color, which is computed directly from the molecular coordinates. Based on all the measured pair-wise distances the algorithm then automatically generates the probability distribution of the distances (probably density) using a cross-correlation (or pair-correlation) function. To simplify, this provides a robust statistical measure for the distance distribution that only converge if there is non-random probability whereby a number of molecules of one color are within a certain range (or distribution) of distances from the molecules of another color. Importantly, if molecules

are placed at random distances the probability will be zero even at incredibly high-densities of molecules since it will yield similar frequency for all distances, whereas even a small subset of molecules (within a field of high-density of randomly localized molecules) are positions within a range of distances from one another will yield a non-zero probability. This means that if there are few non-random events (statistically significant) where pairs of molecules are within (or near) a distance of 20 nm or 30 nm of each other, these are considered as being associated with the same process or complexes, even though their specific molecular coordinates might not overlap as these are resolved at 10 nm. We therefore used the term “NEAR” to indicate changes in the frequency of molecular-pairs that are co-localized to these regions. In our measurements of molecular complexes at replication forks and repair complexes, the range of distances we obtained are less than 100 nm, which also reflect the spread of the SMLM signals including that of EdU at individual forks. Importantly, these distances are far below the diffraction limit of light, and as such will be considered as co-localization of overlapping clusters when analyzed using standard confocal or epi-fluorescence microscopy. We trust that the above descriptions address the reviewers’ questions, with the detailed technical explanations are provided in the cited work and related snippets below. We also provide brief response to the following specific questions:

To how many nanometers distance does this correspond and how has this threshold for being near been set?

See details in the response above and in refs. Briefly, our SMLM analysis utilizes a cross correlation probability density function analysis between two species **that does not rely on a specific distance threshold between two species for consideration of proximity**. This analysis utilizes robust and unbiased computational framework that is based on pair-correlation function where it derives the probability density (or amplitude) as a function of all the pair-wise distances, providing a distance range rather than a finite distance. We emphasize that this sort of calculation does not utilize any distance threshold or data thresholding. In the present work this analysis revealed that the pair-wise interactions of fork associated events occurred on a scale (or range) of within 100 nm, which is the anticipated molecular scale that would encompass such events.

Can one come up with a combined proxy for sites near Pol II and EdU positive regions?

We have complemented our SMLM approach with PLA in multiple instances throughout the paper, where our results from both techniques show consistent data.

It would also be important to show more of the raw data, for example the total level of EdU incorporation, potential differences in S-phase stages of the cells, etc as such parameters may also bias the results.

We thank the reviewer for this comment. For SMLM analysis we specifically only analyze S-phase cells, as previously described in our work with references above. We agree in certain instances in this work it is important to show total EdU and pSer2 RNAP2 levels by SMLM, particularly regarding the impact of CDC7i on nascent DNA synthesis and transcription as pointed out by this Reviewer in major point #12. We have included total nuclear levels of EdU and pSer2 RNAP2 as part of **new Supplementary Fig. 6b-c**, and addressed below.

7. The authors have performed DRIP-qPCR analysis of well-characterized R-loop genes (Fig S4D). However, only one gene is shown (ACTB). 2-3 additional examples are needed to support the idea that BRCA2 KD increases R-loop levels at the TTS. Moreover, it remains unclear to me if ACTB is indeed one of the genes which shows the dormant origin firing phenotype at the TTS. The authors could take advantage of their OK-Seq dataset and confirm the phenotype on a few additional example “dormant origin firing” genes.

We agree with the Reviewer’s excellent point here. We have now included DRIP-qPCR analyses at 6 candidate genes (chosen based on the origin call analysis showing new origins upon BRCA2 deficiency), where we observed increased R-loop signal by DRIP-qPCR at the TTSs of these genes. This is now part of **new Fig. 5c**.

Rebuttal Figure 4 (Corresponding to new Fig. 5c). DRIP-qPCR analysis of 6 candidate gene TTSs in WT- and KD-BRCA2 FTEs +/-RH1 overexpression. Bar graph shows % of input normalized to control sample (0d+EV) of 3 biological replicates. *p* values calculated using two-way ANOVA with Tukey's multiple comparisons test. Error bars = mean, std

8. BRCA2 KD in combination with additional RNaseH1 OE to rescue R-loop levels would be needed to strengthen the point of BRCA2 deficiency induced R-loops at the TTS. This would also further support the point made in the PLA in Figure 4A.

We believe this is addressed above.

9. Rescue experiments are first executed by RNaseH1 overexpression, followed by RNaseH2 depletion experiments in another cell line (U-2OS). There is no rationale provided by the authors for this rather abrupt transition in the manuscript. Can the authors reproduce their findings in U-2OS cells with RnaseH2 in FTE cells?

As FTEs are derived from primary cells, they have been shown to be extremely sensitive to specific perturbations including siRNA reagents. The following figure demonstrates dramatic cell death with siBRCA2 and siRNaseH2A alone:

Rebuttal Figure 5 (Corresponding to OLD Fig. 4f). Plot of percent survival following cell viability assay of FTEs transfected with siRNAs against BRCA2 and/or RNaseH2A following three days of knockdown. *p* values calculated using student's t-test of three biological replicates. Error bars = mean, std.

We therefore performed these knockdowns in U2OS cells based on their tolerance to these siRNA combinations in comparison to the FTEs. The increase in pSer2 RNAP2 – PCNA PLA signal in U2OS cells following BRCA2 siRNA knockdown showed that loss of BRCA2 in different cells phenocopies the PLA results in FTEs with our doxycycline-inducible BRCA2-KD system.

Rebuttal Figure 6 (Corresponding to OLD Fig. 4b). Left: Whole cell lysates of U2OS cells transfected with siRNAs against BRCA2, RNaseH2A or both siRNAs for 3 days. Right: PLA of pSer2-RNAP2 and PCNA with the indicated siRNAs in U2OS cells. *p* values calculated using Mann-Whitney rank-sum t-test of 200 nuclei per condition from 3 biological replicates. Error bars = mean, std.

However, we have decided to remove these two experiments pertaining to the loss of both BRCA2 and RNase H2A from the manuscript as we feel we have addressed the cooperation of this complex in stronger ways, discussed more in major point #11. Furthermore, we feel that addressing synthetic lethal mechanism/function detracts from the point of our study. Future studies from our group will continue to explore the molecular mechanisms of this complex and how they cooperate together to alleviate toxic R-loops at TTSs of active, long genes.

10. Figure 4D – again as a non-expert in this technique: How close does RNH2 need to be to be considered “near”? How do the results change upon changing that distance parameter?

We hope we have addressed this in response to major point #6 above.

11. The authors propose that BRCA2 and RnaseH2 functionally cooperate to mitigate HO TRCs. This is a very interesting

idea and supported by the synthetic lethality. Do both proteins actually interact? Can this be validated by e.g Co-IP Western blot analysis? Related to this, the authors can also check for a potential interaction of BRCA2 with RNaseH1. Thank you for this suggestion. Previously published work showed BRCA2 co-IPs with RNASEH2A in both untreated and IR-treated 293T cells, specifically BRC repeats 1 and 3 of BRCA2 in recombinant purified assays (Fig. 6 in PMID: 30560944). To address the interaction between BRCA2 and RNASEH2A in our model, we performed PLA between these two target in untreated, HU-treated, and ATRi-treated cells. PLA between BRCA2-RNASEH2A shows increased signal upon HU treatment and ATRi. Interestingly, using PLA, BRCA2 appears to also interact with RNase H1 as well upon HU treatment. These data are included as **new Fig. 4h-i** and included below.

Rebuttal Figure 7 (Corresponding to new Fig. 4h-i).

(h): PLA of BRCA2 and RNASEH2A or BRCA2 and RNase H1 in FTEs +/- 0.2mM HU for 4h, including single antibody controls for each PLA. Data presented shows 200 nuclei from 3 biological replicates. *p* values calculated using unpaired two-tailed t-tests between conditions. Left: number of foci per nucleus. Right: average number of foci per nucleus per biological replicates. Error bars = mean, std. Scale bar = 10µm in representative pictures.

(i): PLA of BRCA2 and RNaseH2A or BRCA2 in FTEs +/- ATRi. Data presented shows 200 nuclei from 3 biological replicates. *p* values calculated using unpaired two-tailed t-tests between conditions. Left: number of foci per nucleus. Right: average number of foci per nucleus per biological replicates. Error bars = mean, std.

12. It would be important to show that global DNA synthesis or total levels of Ser2P/Ser5P are not affected by the used concentration of CDC7i. Moreover, I disagree with the assumption that CDC7i will exclusively inhibit dormant origins, rather new origin firing in general. This should be revised in the manuscript.

We have based this assumption on the fact that CDC7i did not decrease PLA signal between pSer5 RNAP2 and PCNA in BRCA2-WT or BRCA2-KD FTEs (**new Fig. 3e**). We believe pSer5 RNAP2 – PCNA signal could be a proxy for origin firing at canonical origins.

Due to the suggestion to perform SMLM on BRCA2-deficient cells +/- CDC7i (discussed more in major point #14), we have included this analysis and raw data below which we believe will address these valid concerns on the potential pleiotropic effects of CDC7i. Below is a representative STORM image of BRCA2-WT and -KD FTEs +/- CDC7i, pulse-labeled with EdU for nascent forks, and stained for pSer2 RNAP2 and gH2ax for SMLM. Previous work from our group utilized SMLM and downstream auto-correlation analysis to measure EdU density per nucleus, as well as the average density of labeled targets of interest (PMID: 35365626). The histograms below show the average EdU densities (**B**) and average pSer2 RNAP2 densities (**C**) per nucleus of two biological replicates. Our results show that neither loss of BRCA2 nor inhibition of CDC7 in either BRCA2-WT or -KD cells impact overall EdU-incorporated nascent synthesis in S-phase nuclei (**B**). This is also consistent with previously published work from our group showing CDC7i did not significantly impact origin firing in ATR-proficient cells (PMID: 34473946).

CDC7i shows decreased global pSer2 RNAP2 in BRCA2-WT cells (**C**), likely due to simultaneous off-target CDK9 inhibition by this drug, which has been previously reported (PMID: 18469809). Loss of BRCA2 in FTEs decreases global nuclear pSer2 RNAP2 abundance by STORM (**C**). We have other evidence that loss of BRCA2 globally negatively impacts transcription by nascent EU-IF and ChIP-seq (data not shown, ongoing work in our group). These results are also potentially consistent with one previously described role for BRCA2 promoting transcription elongation at highly-transcribed genes (PMID: 29386125). Importantly, CDC7i does not further decrease global pSer2 RNAP2 levels in BRCA2-deficient cells (**C**), suggesting the decreased TRCs seen by PLA in BRCA2-KD (**new Fig. 3d**), and decreased gH2AX near pSer2 RNAP2 by PLA and SMLM (**new Fig. 3i**, **new Fig. 3j**) are likely not due to CDC7i impacting overall pSer2 RNAP2 levels or global DNA synthesis when BRCA2 is knocked down.

This new result testing the effects of CDC7i on global DNA synthesis or total Ser2P/Ser5P levels in FTEs is now part of **new Supplementary Fig. 6b-c**.

Rebuttal Figure 8 (Corresponding to new Supplementary Fig. 5b-c).

(b): Representative SMLM image shows EdU (yellow) of pulse-labeled FTEs +/- shBRCA2 +/- CDC7i. Scatterplot quantification measuring total EdU levels per nucleus. *p* values calculated using unpaired two-tailed t-tests of at least two biological replicates: WT-BRCA2(-)CDC7i, N=91; WT-BRCA2(+)-CDC7i, N=96; KD-BRCA2(-)CDC7i, N=83; KD-BRCA2(+)-CDC7i, N=75;. Error bars = mean, std.

(c): Representative SMLM image shows pSer2 RNAP2 stain (cyan) of FTEs +/- shBRCA2 +/- CDC7i. Scatterplot quantification measuring total EdU levels per nucleus. *p* values calculated using unpaired two-tailed t-tests of at least two biological replicates: WT-BRCA2(-)CDC7i, N=91; WT-BRCA2(+)-CDC7i, N=94; KD-BRCA2(-)CDC7i, N=83; KD-BRCA2(+)-CDC7i, N=75;. Error bars = mean, std.

13. ATRi inhibition also increases the TRC PLA foci of Ser5P-PCNA (Fig S4F). It is difficult for me to conclude from this that ATRi necessarily produces CD TRCs due to more firing at canonical origins. ATRi-driven HO TRCs seen with Ser5P-PCNA could also stem from cryptic intragenic origins and HO TRCs. Since it remains technically impossible to determine from what genomic location a PLA focus originated from, I would suggest that the concept that Ser2P equals HO TRC and Ser5P equals CD TRCs should be more carefully discussed throughout the manuscript.

Thank you for this note, we agree that we can't rule out that ATRi could force cryptic intragenic origins, we now provide a more careful discussion of the Ser2P vs Ser5P points in regards to either HO or CD TRCs in the discussion (see page 21).

14. The proximity analysis between Pol II Ser2P and various DNA damage markers with and without CDC7i is a strong point of the story. It would be nice to validate a few of those with SMLM to ensure the damage localizes to the TRC region directly.

Thank you for this suggestion. We agree it is a strong result by PLA and we validated this result by SMLM, which is now incorporated as **new Fig. 3g, i**.

Minor points:

- In the abstract the authors mention that "BRCA2 is a tumor suppressor protein responsible for safeguarding the cellular genome from replication stress and genotoxicity, but the mechanism(s) by which this is achieved remains elusive." This should be rephrased as there is a significant amount of literature on the multiple mechanisms how BRCA2 safeguards the genome, e.g. its function to bind to RAD51 to promote HR, as also mentioned by the authors in the limitations of the study.

Thank you, we have rephrased this sentence in the text.

- In the introduction (line 8) the authors state that endogenous sources of DNA damage drive genomic instability and cancer progression in FTEs despite them being unclear. Is this known from literature before this study? Please cite or rephrase.

High levels of gH2AX in secretory epithelial cells of the fallopian tube fimbriae characterize early HGSOC development (PMID: 17117391). This has been made clearer in the introduction.

- Figure 1A, it would help the readability of the figure to provide an X-axis labels on the TSS to TTS relative distance plot to make it more easy to grasp what is shown here.

Thank you for the suggestion. We have expanded our explanation of the relative distance plots in **new Fig. 1a**, in the main text, and in the methods.

- Figures S1G/H would profit from figures titles marking them as high/low volume genes

We agree and have added titles to the figures for clarity.

- On page 7 in line 33 the authors state that Ser5P phosphorylation of the CTD occurs to promote imitation and promoter proximal pausing a few kilobases downstream of the TSS. This statement needs to be revised. First, promoter proximal pausing occurs 50-70 bp downstream of the TSS not kilobases. Moreover, the Ser5P mark is to my knowledge

highly present at paused Pol II but not directly implied in promoting pausing in comparison to elongation factors like DSIF or NELF.

We have updated this sentence with more careful language.

- On page 8 line 2 the authors say that Ser2P Pol II accumulates downstream of gene bodies. This should also be rephrased as Pol II Ser2P starts accumulating downstream of the pausing site throughout the gene body with highest levels before the TTS. In my view, downstream of the gene body would be outside of the gene already.

We agree this is unclear and have updated our language in this section.

- The authors state “Considering intragenic origin firing is suppressed in actively transcribed genes (Figures S2B-F), the proximity between various phospho-forms of RNAP2 and active forks arising from intergenic origins marks distinct TSS- and TTS-oriented locations for TRC events”. This statement should be rephrased as an hypothesis and not considered a fact since no existing literature has convincingly proven this assumption yet.

We have updated our discussion of this more carefully.

- For the PLA experiments in Figure 3C statistics should be provided like in all other PLA experiments.

The statistics for this figure have been updated and are now part of Fig. 4c.

- Figure 5B and 5C - Quality of track images/contrast could be improved.

We have adjusted the contrast of the images to improve on the quality of the fiber tract images in new Fig. 3b-c.

- In the discussion the authors speculate that reduced Ser5P-PCNA interaction in PLA could be stemming from decreased canonical origin usage upon BRCA2 deficiency. With their Ok-seq data set at hand the authors are uniquely positioned to address this question. This analysis would provide crucial insights into a shift from CD to HO conflicts at highly transcribed genes upon BRCA2 loss.

Thank you for this suggestion. Indeed, Ok-seq analysis of replication fork directionality (RFD, proportion of forks moving left to right) followed by metagene analysis of genes allows for sensitive detection of origin efficiency or delocalization. Sample data below shows how changes in these parameters are displayed by this particular Ok-seq analysis.

[redacted]

Rebuttal Figure 9. Adapted from Liu et al. (PMID: 33442052). Schematic of Ok-seq data showing percent of forks moving left to right oriented around meta-origin. Strong Ok-seq origin firing at TSSs shows increased percent of forks at TSS, while delocalized origin firing shows increased percent of forks upstream or downstream of TSS.

The following data is OK-seq RFD by metagene analysis of gene FPKM by quartiles, where the top quartile represents the most highly transcribed genes, and the bottom quartile represents the least transcribed genes based on RNA-seq of FTEs. % of forks moving left>right mapped around TSSs and TTSSs of these data sets. Indeed, RFD analysis by Ok-seq shows less efficient origin firing, without origin delocalization, at the most highly-transcribed genes in BRCA2-KD in FTEs. However, ongoing work in our lab is exploring specifically decreased origin firing at TSSs in BRCA2-KD (see below), therefore we are hesitant to include this data in this manuscript at this time.

Rebuttal Figure 10. Percent of forks moving left to right around TSSs binned by total RNA-seq read depth quartile (most highly transcribed gene TSSs at top, least transcribed gene TSS quartile at the bottom) of BRCA2-WT FTEs (black) and BRCA2-KD FTEs (red).

Reviewer #2 (Remarks to the Author):

Goehring et al use Ok-seq, PLA and super resolution microscopy to study the effects of BRCA2 depletion on origin firing and transcription-replication collisions (TRCs) in fallopian tube epithelium (FTE) cells, the likely cell-of-origin for high-grade serous ovarian carcinoma (HGSOC). They propose that forks originating from TSSs of transcriptionally high-volume genes are more inclined to slow-down or stall in BRCA2-deficient cells, resulting in increased downstream origin firing and increased head-on (HO) TRCs in the 3' end of these genes. They propose that BRCA2 can directly mitigate R-loops at HO-TRCs through the recruitment of RNaseH2, and indirectly suppress HO-TRCs through the passive inhibition of replication stress-induced dormant origin firing. They find that that endogenous sites of HO-TRCs are enriched for gamma-H2AX, phosphorylated RPA, and R-loop accumulation, which can be mitigated by limiting dormant origin firing with inhibitors of the origin firing kinase Cdc7. Finally, a bioinformatic analysis suggests that the affected genes are enriched in several oncogenic pathways including the epithelial-mesenchymal transition, suggesting possible pathways for HGSOC progression. Overall, this is an impressive study rich in novel data. However, I found that several experiments were insufficiently explained. Furthermore, the proposal that BRCA2-depletion results in head-on rather than codirectional collisions is in my opinion insufficiently demonstrated.

We thank this reviewer for their encouragement and helpful suggestions. We have expanded our explanations to further the understanding of this study.

1. The description of the "new computational methodology to analyze our previous Ok-seq dataset to probe for replication origins from Ok-seq reads mapped to the Watson and Crick strands" is complex and not sufficiently clear and illustrated (page 5, lines 17-21, Fig 1 A, Fig S1A). The authors compute increments in Crick or Watson Okazaki fragment abundance between consecutive 1-kb windows, compute the log10 of the inverse product of these two measurements, smooth the results using a Hann function with a 60 kb window and filter out the origins whose score is less than the

lower bound of the full width at half maximum (FWHM) of the score distribution. This was tentatively illustrated in Figure 1A and S1A but in a too simplistic manner (e.g. the Hann smoothing is not explained).

First, the authors should illustrate each of these steps for a large chromosomal segment (say, one or a few Mb) containing large transcribed genes whose origin distribution is affected by hydroxyurea treatment. They should plot i) the Watson and Crick Okazaki fragment abundance profiles in the absence or presence of hydroxyurea; ii) the replication fork directionality ($RFD = (C-W) / (C+ W)$) as in previous publications; iii) the unsmoothed discrete derivatives by 1 kb windows $DER(Watson)$ and $DER(Crick)$; iv) the Hann-smoothed profile(s); and v) the position of origins retained after filtering. Thus, the readers will be able to judge by themselves, on this exemplary locus, the magnitude of the effects of hydroxyurea, from raw to processed data, and to connect the origin scores with the actual changes in Okazaki fragment abundance and RFD profiles.

Second, I anticipate that changes in Okazaki fragment abundance in successive 1-kb windows predominantly reflect statistical noise in read counts rather than actual changes in fork direction. The derivative of a noisy signal being unstable and not meaningful, the Hann smoothing step is crucial because it may affect the precision of mapping origin activity changes. In short, do the observed score changes reflect global changes over a 60 kb window or can they be confidently attributed to narrower (1-kb ?) windows ?

We thank the reviewer for this very important question and opportunity to clarify our Ok-seq origin analysis. We would first like to clarify that the Hann (hanning) smoothing using a filter of width 60kb is applied to the raw 1kb-binned Crick/Watson signals, before the discrete derivative of each W/C array is calculated to generate an origin score. This was not clearly written in our methods and we have updated this. We have also updated our schematic in Figure 1 to include mention of this, as well as our explanation in the text. We thank the reviewer for pointing out this opportunity for confusion.

The Hann smoothing is applied to reduce statistical noise prior to taking the derivative of the W/C signals. The exact filter we used is `np.hanning` (<https://numpy.org/doc/stable/reference/generated/numpy.hanning.html>). We agree with the reviewer that the smoothing filter size would definitely impact signal/noise of the W/C read density and therefore origin call analysis. We settled on a filter width of 60kb as it provided a good overlap between replicates in our originally published Ok-seq RPE data (PMID: 30598550). The figure below shows the % of unmatched origin calls between replicates in untreated RPE cells (y-axis) at increasing levels of Origin Efficiency cutoff (x-axis), for origin calls at Hann smoothing windows of 20-70kb (legend). Origin Efficiency is measured as the difference of the ratio $C/(W+C)$ to the right and left of the origin midpoint (modified version of the OEM from PMID 23562327).

Rebuttal Figure 11. Percent of unmatched origin calls between replicates from untreated RPE Ok-seq sample (PMID: 30598550) at various levels of efficiency cutoffs. Efficiency cutoff is defined as $c/(w+c)$ averaged around a 20kb window to the right and left of the origin midpoint.

While a narrower Hann filter may allow us to detect more putative origins, we opted to prioritize a filter size that would reduce the false positive rate and maximize origin call consistency between replicates. The plot above illustrates that origin calls with a narrower smoothing window (<50kb) result in a relatively high disagreement between replicates. These would be difficult to analyze since changes at this lower resolution would be hard to distinguish from the statistical noise inherent in the data. Since the same filter is uniformly applied to distinct Ok-seq biological replicates and between conditions, we have confidence in our ability to discern the nature of the origin changes in HU-treatment or BRCA2-KD.

Finally, we thank the reviewer for the excellent suggestion to show more visual Ok-seq raw/processed data in helping the reader understand the origin call analysis. Below we have provided in a 1-MB window for each Ok-seq sample of untreated RPEs and HU-treated RPEs from previously published work (PMID: 30598550) the following tracts at an exemplary locus:

A. W and C Ok-seq abundance profiles

- B. Hann smoothed W and C reads
- C. RFD
- D. Derivative origin scores
- E. Origins retained for downstream analysis

We did not include “iii) the unsmoothed discrete derivatives by 1 kb windows DER(Watson) and DER(Crick)” as the reviewer suggested, because (as hopefully clarified above) our analysis does not rely on taking the derivative prior to smoothing. We apologize again for this confusion and we hope this visualization provides clarity on our origin call analysis. This important visualization has been incorporated as **new Supplementary Fig. 1d**.

Rebuttal Figure 12 (corresponding to new Supplemental Fig. 1d). Origin call analysis pipeline visualization. Representative locus showing (i) 1-kb binned Watson and Crick Ok-seq read abundance profiles, (ii) Hann smoothed Watson and Crick reads, (iii) replication fork directionality calculated as $(C-W)/(C+W)$, (iv) DER-SCORE calculated from increasing DER(Crick) and decreasing DER(Watson), (v) origin calls retained after normalization and FWHM filter, each for untreated RPE Ok-seq sample (black) and HU-treated RPE Ok-seq sample (blue). hg19 refseq genes shown.

2. The "relative distance calculation" procedure used to decide whether origins map at TSSs or near TSSs or are "not oriented around TSSs" was not sufficiently explained (page 5, line 25). Why use a relative rather than an absolute distance measurement? Relative to what? To gene length? How is it computed? Why not simply provide the absolute distance between origins and the nearest active (or inactive) TSS? In Fig 1B,C,F,G, and many similar Figures throughout the paper, is it the absolute value of the distance or the arithmetic distance that is plotted from 0 to 0.4? In other words, do these Figures only show the distribution of origins inside the gene, or a superimposition of origins inside and outside the gene? In the former case why only show intragenic origins? In the latter case it would be preferable to separately plot extragenic and intragenic origins using positive and negative relative distances rather than their absolute values. Please clarify.

Thank you for the opportunity to clarify. The relative rather than absolute distance calculation was used because overall the majority of all of the origins from our analysis are relatively enriched at TSSs and intergenic regions (**Fig. 1e**). Based on this data, we originally tried calculating the absolute distance of the origins in each replicate from TSSs of high-volume genes ($FPKM \times \text{length} > \text{median}$), which showed no differences between BRCA2-WT and -KD. The graph below shows the \log_2 -normalized absolute distances of all Ok-seq origins in each replicate at 0 or 3 days of shBRCA2 in FTEs from each high-volume TTS. The *bedtools closest* function was used to calculate these distances in kilobases (<https://bedtools.readthedocs.io/en/latest/content/tools/closest.html>). Unpaired t-tests between BRCA2-WT and -KD of each replicate reveals no difference in absolute distance, suggesting this approach does not allow us to capture the

trend of data we observed in Fig.1e. Because so many origins are enriched at likely large, intergenic regions of the genome, the new trend of TTS-oriented origins seen in Fig.1e is likely drowned out by these origins, as well as the canonical TSS-origins that don't change in BRCA2-KD.

Rebuttal Figure 13. All replication origins identified by OK-seq origin call analysis from two biological replicates of FTEs showing log-normalized distance to nearest TTS.

On the other hand, the relative distance calculation allows for comparison of loci genome-wide that relies on spatial correlation rather than an absolute value which might be drowned out by the magnitude of the genome or sparsity of TTS-origins (in this case due to increased correlation at other features like TSS/intergenic regions). This relative distance calculation allows us to focus on the global orientation of Ok-seq origins near active TTSs as they appear throughout the genome, and we feel that by comparing BRCA2-WT and -KD Ok-seq replicates, we achieved consistent data to other global mapping annotation (**Fig.1e**). More information on the relative distance calculation can be found here: <https://bedtools.readthedocs.io/en/latest/content/tools/reldist.html> and in the original paper describing the calculation used for genomic annotation (PMID: 22693437). We have updated main **Fig. 1a** (below) and the corresponding text to improve the explanation of the relative distance calculation.

Rebuttal Figure 14 (Corresponding to Fig. 1a). Identification of new origins at TTSs upon replication stress and schematic of Okazaki-fragment sequencing origin analysis. (i) 1-kb binned Crick and Watson Ok-seq reads are Hann smoothed and replication origins calls are defined as regions of increasing Crick-strand derivative and decreasing Watson-strand derivative from smoothed reads (see Methods). (ii) Relative distance calculation orients origins to genomic annotations (TSS, TTS). Replications origins are prevalent at active TSSs indicated by high proportion of origins at low relative distances active TSSs (left). If origins were randomly distributed with respect to active TSSs, there would be an equal proportion at all relative distances (right).

All relative distance calculations used all of the origins called from our origin analysis, and we did not filter for intragenic or extra genic origins prior to analysis. Therefore, this analysis shows a superimposition of origins that could be found both inside and outside of a gene. As this reviewer suggested, we performed the relative distance calculation with both intergenic and extragenic origins, relative to the nearest active TTS (gene volume (FPKM x length) > median). The data below shows that intergenic origins oriented near TTSs of high-volume genes are statistically significantly increased in BRCA2-KD compared with BRCA2-WT (left). However, genic origins are strongly disoriented away from the TTSs of high-volume genes, with no statistically significant impact of BRCA2-KD (right). This is consistent with a similar analysis presented in **Supplementary Fig. 3d**, showing that the majority of origins that are genic are in very low transcription genes. We thank the reviewer for this important point that we feel improves our story. The data below has been incorporated as new **Supplement Fig. 3b-c**.

Rebuttal Figure 15 (Corresponding to Supplement Fig. 3b-c). Relative distance analysis of FTEs +/- shBRCA2 of either intergenic origins (left) or intragenic origins (right) relative to high-volume gene TTSs.

3. The authors state that there is "a striking increase in origins oriented at TTSs of high-volume genes compared to untreated conditions (Figure 1C)." However, looking at the graduations of the y-axis, this increase is only from 2.2% to 3.2% of origins mapping at the TTS. While this seems significant, it is only a weak effect, somewhat visually amplified because the y-axis does not start at 0. In Figure 1G the increase in origins at TTSs following BRCA2 depletion seems even smaller (from 2.0% to 2.3%). Please replace "striking" by an objective quantitative measurement.

We have replaced "striking" with "significant".

4. I could not find the Extended Supplement Data with exact distances between TTSs and new origins in BRCA2-KD that is said to support the identification of candidate genes with new origins near TTSs (page 6, line 27).

Thank you for pointing this out. We have added this data in **Source Data (excel file)**.

5. Figure 1i, right panel, shows the appearance of new intragenic origins following BRCA2-KD, in apparent contrast with the statement that "BRCA2-KD did not impact levels of intragenic origin usage (Figure 1E)" (page 7, lines 32-34). Please explain.

The gene shown in Fig.1i (right) is showing new BRCA2-deficient TTS-oriented origins between two genes in tandem (*PHLPP2* on the left and *AP1G1* on the right). We have blown up Fig. 1e below to show this more clearly.

We have now replaced the righthand graph with a clearer representative gene which can also be seen below (in **new Fig. 1i**):

TLE3 - chr15:70,340,129-70,390,256

Rebuttal Figure 16 (Corresponding to Fig. 1i). Representative candidate gene *TLE3* with new origins at TTSs with BRCA2-KD in two Ok-seq biological replicates each with 0, 3, and 6 days of knockdown.

6. Page 8, line 11, reads "Interestingly, pSer5-RNAP2 proximity to PCNA decreased in BRCA2-KD (Figure 2C), consistent with a previous study that found BRCA2-deficient cells have decreased pSer5-RNAP2 signal at promoters of transcriptionally-active genes (ref 23)". However, Reference 23 shows that BRCA2 inactivation by depletion or cancer-causing mutations triggers RNAPII accumulation at promoter-proximal pausing (PPP) sites in actively transcribed genes (Fig 3A in Ref 23). It is not clear if this accumulated RNAPII is phosphorylated on Ser 5, Ser 2, or both, and the small

decrease in the Ser5 form seen on their Figure 3B at PPP does not appear statistically significant, in contrast to total RNAPII accumulation. Please clarify, or remove the part of the sentence "consistent with a previous study..."

Thank you for this comment. We have removed the latter part of that sentence.

7. Figure 2E,F, please clarify whether what is shown is a nuclear slice (in this case, which thickness ?) or the projection of the total nuclear signal on a plane. This is important to evaluate the significance of overlapping signals. Please also clarify what is the unit on the y-axis. I wondered whether the total RNAP2 signal associated with EdU tracks would increase, decrease or remain constant following depletion of BRCA2. Can the authors perform their assay with their pan-RNAP2 antibody ? Finally, the length of DNA labelled during a 10 min EdU pulse should be about 10-20 kb, but shorter if replication was stalled for a few minutes. I would expect RNAP2 to be localized at the border of (on average, shorter) EdU tracks in the case of stalling collision, but anywhere along (full-length) EdU tracks in the case of a moving RNAP2 transcribing a newly replicated DNA stretch without stalling. Does the resolution of the technique allow the authors to discriminate these two scenarios? In other words, do coincident EdU and RNAP2 signals overlap differently for pSer5 and pSer2 forms of RNAP2, and are EdU signals that overlap with RNAP2 less intense than EdU signals that do not overlap with RNAP2?

For our measurements we used established assays described in previous publications (for example, see PMID: 34473946, Supplemental information, Methods S3-S4), and applied the same experimental parameters. We added references and text to the methods section that clarified on this point. Briefly, for SMLM/STORM all Samples were illuminated in HILO mode which generates a quasi-lightsheet that excites a thin slice (300nm) in the sample plane thereby resulting in enhances out-of-plane rejection ensuring that any observed signals are obtained from the same plane. Further out-of-focus signal is rejected through the SMLM localization algorithm (PMID: 34473946). In-depth description of the technical parameters of the methods, experimental assays and analysis are provided in PMID: 34473946, Supplemental information, Methods S3-S4 as well as in our other work (for example in PMID: 33953191, PMID: 33370257, PMID: 32542039, PMID: 31570834, PMID: 30631072, PMID: 30422114, PMID: 30250272)

The unit on the y-axis is the average density of species A near species B based on our cross-correlation analysis previously described (PMIDs: 25843623, 34473946, 30422114, 30250272, 35365626) per μm . We have updated the Y-axes on the corresponding figures with this unit. A detailed explanation of the cross-correlation analysis can also be found in the response to Reviewer 1, major point #6.

We have found strong agreement between our STORM and PLA data. PLA probing for TRCs using pan-RNAP2 and PCNA showed no difference between BRCA2-WT and -KD cells (Fig. 2d), likely due to the fact that the pan-RNAP2 is detecting multiple phosphoforms of RNAP2 and provides the least information on RNAP2 status.

Finally, evaluating the density of EdU tracks that are coincident with pSer5 RNAP2 vs pSer2 RNAP2 is also a strong suggestion that could allow us to make inferences on whether the TRCs are HO or CD. Unfortunately, this is not possible by our STORM analysis due to the difference in staining between the two phosphoforms of RNAP2; it would be difficult to compare overlapping signals of each unique antibody compared to EdU with confidence. Further, even if we were able to perform this experiment, we suspect evaluating whether less intense EdU signals as a proxy for stalled forks would still not give us much information on the orientation of TRCs. The increased PLA signal between pSer33-RPA and pSer2-RNAP2 in BRCA2-KD (new Fig. 3k) suggests the forks near pSer2-RNAP2 are stalled. We have other data that suggests that canonical forks emanating from TSSs are also stalled in BRCA2-KD (data not shown). We therefore hypothesize that both TSS-oriented canonical forks and TTS-oriented forks from new origins are stalled in BRCA2-loss. However, ongoing work in our group is pursuing using new tools to track the directionality of nascent DNA/RNA tracts with dual labeling probes to address this question more thoroughly, but this is beyond the scope of the current study.

8. Page 9, lines 1-4. The interpretation that forks generated from TTS origins converge with elongating RNAP2 to increase HO-TRCs is likely but not proven. PLA and SMLM experiments suggest that BRCA2 depletion increases colocalisation of RNAP2 with PCNA and with newly synthesized DNA, but do not discriminate between head-on and co-directional collisions. One could for example argue that a significant fraction, or even a majority, of these colocalisation events correspond to co-directional collision of forks emitted from the TSS, and that a consequent fork slowing would give more time for origins at and downstream of the TTS to fire and rescue the stalled co-directional forks. Can the

authors formally exclude this (or other alternative hypotheses) and give more convincing evidence for head-on collisions ? Same criticism for the interpretation of the effect of hydroxyurea, Fig S4A, page 9, lines 10-12.

It is possible that some of the collisions observed in BRCA2-KD are due to codirectional collisions from TSS-forks due to either forks slowing or transcription dysregulation at these regions. However, we suspect in this scenario, we would have also observed increased pSer5 RNAP2 - PCNA PLA signal in this case or increased pSer5 RNAP2 – EdU pair correlation analysis by STORM, which we do not see (Fig. 2c, e).

We performed DRIP-qPCR at candidate genes identified by Okseq to evaluate R-loops levels in BRCA2-WT and -KD conditions, +/- overexpression of RNase H1. These results show that we observe R-loop accumulation in BRCA2-loss, and this signal is sensitive to RNase H1 overexpression (new Fig. 5c):

Rebuttal Figure 4 (Corresponding to new Fig. 5c). DRIP-qPCR analysis of 6 candidate gene TTSs in WT- and KD-BRCA2 FTEs +/-RH1 overexpression. Bar graph shows % of input normalized to control sample (0d+EV) of 3 biological replicates. *p* values calculated using two-way ANOVA with Tukey's multiple comparisons test. Error bars = mean, std

Interestingly, DRIP-qPCR analysis does not show BRCA2-deficient R-loop accumulation at the TSS or introns at three of these candidate genes (presented in new Supplementary Fig. 7a):

Rebuttal Figure 17 (Corresponding to new Supplementary Fig. 7a). DRIP-qPCR analysis at TSSs and introns of 3 candidate genes in WT- and KD-BRCA2 FTEs +/-RH1 overexpression. Bar graph shows % of input normalized to control sample (0d+EV) of 3 biological replicates. *p* values calculated using two-way ANOVA with Tukey's multiple comparisons test. Error bars = mean, std.

While not all TRCs are associated with R-loop accumulation, and the specific dynamics of how CD- or HO-TRCs might impact R-loops levels are still unclear, we hope these experiments shed light on the possibility that these TRCs are enriched at TSSs. We suspect CD-TRCs would show an increased R-loop signal at TSSs and introns of TRC prone genes in our DRIP-qPCR experiment, but we did not observe this effect.

Additionally, we thank this reviewer for the comments raised in major point #2 which have allowed us to add to our relative distance analysis of Ok-seq origins. As discussed above in major point #2, the increased origins in BRCA2-KD are enriched for intergenic origins. Genic origins in BRCA2-WT and -KD Ok replicates are not oriented around TSSs of high-volume genes, consistent with our data presented in new Supplementary Fig. 3c. Therefore, because of the orientation of increased BRCA2-KD origins at TSSs or downstream of TSSs (by nature of being intergenic) (new Supplementary Fig. 3b), the orientation of these origins relative to high-volume TSSs gives us more confidence that these interactions are likely in the HO orientation.

9. Page 9, lines 20-22, " the PLA signal between BRCA2 and PCNA were increased upon HU treatment, revealing the elevated recruitment of BRCA2 to active forks" : did they mean stalled forks ?

Thank you for allowing us to clarify. For the experiments in new Fig. 4, we treated cells with a low-dose (0.2mM) of HU

for 4 hours. Previous work from our lab and others has shown that this low-dose HU condition generally slows replication forks (very few stalled/collapsed forks) but also induce firing of new origins/forks (PMID: 25843623, 18079179). Based on the SMLM data in the **new Fig. 4b** (middle), upon HU treatment there is increased BRCA2 recruitment to nascent DNA (EdU). PLA in these conditions similarly shows increased BRCA2/PCNA colocalization upon low-dose HU treatment (**new Fig. 4c**). Based on the SMLM data in Figure 4, at least a subset of BRCA2 recruitment to forks are to new forks in this low-dose HU setting (**new Fig.4B**, middle).

10. DNA fiber analyses in Fig 5. It has been shown that measurements of IODs and fork speeds can be strongly biased by the length of DNA fibers analyzed (Técher et al PMID: 23557832). It is therefore important to show that fiber size distribution is similar in different samples to allow meaningful comparisons. This important control was apparently not performed.

Thank you for alerting us to the Techer et al study looking at lengths of DNA fibers. DNA fiber analysis in our lab is quite routine and we generally don't see dramatic variations with overall DNA fiber lengths from experiment to experiment using the tilt-method of DNA fiber stretching. In previous experiments, we have confirmed this using yoyo-1 staining for DNA fiber quality, density, and lengths. Since our IOD and fork speed measurements are in line with published literature, we do not suspect any major effects on tract lengths is due to experiment-to-experiment variability.

Reviewer #3 (Remarks to the Author):

In this manuscript, the authors interrogate whether and how replication stress promote genome instability and cancer progression. They do so in the context of BRCA2 deficiency, in a relevant model of high-grade serous ovarian carcinoma (HGSOC), of which a good fraction is homologous recombination deficient (HRD).

Given the role of BRCA2 in preventing or suppressing R-loops previously established, and in particular the role of BRCA2 in recruiting RNaseH2 to DNA-RNA hybrids at sites of DSBs, and previous report on the regulation of transcription by BRCA2, they investigate whether BRCA2 deficiency affects replication fork dynamics and in particular, transcription-replication conflicts that could explain the genome instability that fuels tumorigenesis in this setting.

They found that BRCA2 KD leads to firing of dormant origins specifically at TTS of highly expressed long genes, "high-volume genes". Then they tested whether the increased of origin firing lead to the presence of HO-TRCs using the proximity of initiation or elongating RNA pol II and PCNA as a proxy which can they show can be alleviated by PARPi or CDC7i. Finally, they show that the recruitment of RNaseH2 to HO-TRCs to resolve R-loops at HO-TRCs is dependent on BRCA2 implicating the latter in this function.

Overall this an interesting manuscript showing that dormant origin firing induce replication stress in BRCA2 deficient cells ultimately resulting in HO-TRCs at long highly transcribed genes although the relation to RNaseH2 is unclear. The authors further suggest this is particularly relevant in HGSOC oncogenesis for which they provide certain evidence based on patient data.

I think this manuscript has potential but would require major revision for publication, see specific comments below:

We thank this reviewer for the encouragement.

Major:

1. The fact that elongating RNAPol II colocalize with EdU or PCNA is indicative of head-on replication-transcription collisions seems far-fetched. A minimum control in these assays (Fig. 2) would be to test whether RNaseH treatment reduces this colocalization in PLA and SMLM experiments the way they did then in Fig. 4A. Moreover, if the idea is that RNaseH2 deals with these conflicts, RNaseH2 overexpression should be employed to control for this and confirm this is the case.

Thank you for this suggestion. Between three major experiments presented in **Fig. 2e-f**, **new Fig. 3h-i**, and **new Fig. 4b**, we have found strong agreement between our PLA and SMLM results. Therefore, it is likely that RNase H1 overexpression in BRCA2-KD scenario shown in **new Fig. 5b**, would be recapitulated for SMLM. Additionally, we have

provided new data showing that overexpression of RNase H1 in BRCA2-deficient cells reduces R-loops at candidate genes with new origins at their TTS, and therefore potential sites of HO-TRCs (**new Fig. 5c**). This data has been included in response to major point #2 below.

Human RNase H2 is a heterotrimer and, to the best of our knowledge, there has been no precedent for transfection of all three subunits of RNASEH2A, B, and C to reduce / prevent cotranscriptional R-loops. Additionally, we have found that the human fallopian tube epithelial cells (FTEs) that we are working with show increased cell death with high amounts of transfection reagents, and suspect that transfection of the heterotrimer RNase H2 would be challenging in this cell model. A recent, foundational work characterizing RNase H2's role in preventing cotranscriptional R-loops in human cells also employed RNase H1 overexpression for R-loop depletion (PMID: 35618715). We hope our experiments employing knockdown of the catalytic subunit of RNASEH2A (**new Fig. 4G**) and localization of RNASEH2A by SMLM is convincing (**new Fig. 4d-f**). In addition, we provide new data showing that BRCA2 also colocalizes with RNase H1 upon induction of TRCs by PLA (**Fig. 4h-i**). Thus, it is likely that both RNase H2 and RNase H1 play critical roles in mitigating HO-TRCs in BRCA2-deficient FTEs.

2. Although both PLA and SMLM seem to go in the same direction, they are based on the same principle. An orthogonal method such as the use of DRIP (DNA-RNA hybrid immunoprecipitation) using primers at TSS vs TTS would be more convincing also to show that these transcription conflicts occur close to TTS and at loci where the authors have mapped the new active origins (Fig. 1I). This experiment has been performed and is somewhat hidden in Fig. S4D however only with a single locus, minimally this should be done in 2-3 to show that is truly general for the genes affected and it does not happen at "low volume genes". Importantly, is the RNaseH1 treatment I Fig. S4D done in the 3d sample or in the 0d sample? It is not clear from the legend nor the figure and it is critical as they need to show that the one occurring at 3d (BRCA2 depletion) is sensitive to RH1 treatment.

Also, validating the proxy with the Serp5 and Serp2 using the proximity between Serp2 and Serp5 with DNA-RNA hybrids would be more convincing.

These experiments are important because all the rest of the manuscript is based on the assumption that what they observe by PLA are HO-TRC.

We thank this reviewer for this great suggestion. We have now performed DRIP-qPCR at 6 candidate genes (curated genes with new origins at TTS in BRCA2-KD based on our OK-seq analysis) in BRCA2-WT and -KD cells with and without overexpression of RNase H1. Indeed, we see R-loop accumulation in BRCA2-loss, and this signal is sensitive to RNase H1 overexpression (presented in **new Fig. 5c**):

Rebuttal Figure 4 (Corresponding to new Fig. 5c). DRIP-qPCR analysis of 6 candidate gene TTSs in WT- and KD-BRCA2 FTEs +/-RH1 overexpression. Bar graph shows % of input normalized to control sample (0d+EV) of 3 biological replicates. *p* values calculated using two-way ANOVA with Tukey's multiple comparisons test. Error bars = mean, std

Importantly, and in agreement with our model, DRIP-qPCR analysis does not show BRCA2-deficient R-loop accumulation at the TSS or introns at three of these candidate genes (presented in **new Supplementary Fig. 7a**):

Rebuttal Figure 17 (Corresponding to new Supplementary Fig. 7a). DRIP-qPCR analysis at TSSs and introns of 3 candidate genes in WT- and KD-BRCA2 FTEs +/-RH1 overexpression. Bar graph shows % of input normalized to control sample (0d+EV) of 3 biological replicates. *p* values calculated using two-way ANOVA with Tukey's multiple comparisons test. Error bars = mean, std.

We also do not observe any R-loop accumulation at the TTSs of the following negative control genes, which are long but have no or very little transcription based on RNA-seq of FTEs (presented in **new Supplementary Fig. 7b**):

Rebuttal Figure 18 (Corresponding to new Supplementary Fig. 7b). DRIP-qPCR analysis at TTSs of 3 low transcriptional volume loci in WT- and KD-BRCA2 FTEs +/-RH1 overexpression. Bar graph shows % of input normalized to control sample (0d+EV) of 3 biological replicates. *p* values calculated using two-way ANOVA with Tukey's multiple comparisons test. Error bars = mean, std.

3. For all PLA experiments, the signal images and quantification of the antibody alone controls should be shown in the same graph, this is important to discard the signal of the the probes for the two antibodies together is not the sum of the background signal coming from both single antibody controls.

Thank you for this suggestion, we have updated our PLA experiments to include the negative control antibodies in each graph.

4. The paragraph explaining Fig. 3C. does not match the results shown in the image and in the graph. First in lane 21, authors state PLA signal for BRCA2-PCNA is increased upon HU (fork stalling) which it has indeed been reported before. However, in lane 25, they state the opposite: "fork-stalling alone is insufficient to promote BRCA2 recruitment at the forks". All of this for a graph where there is no statistical analysis.

In addition to showing the statistics this part needs clarification.

Thank you for allowing us to clarify. For the experiments in **new Fig. 4**, we treated cells with a low-dose (0.2mM) of HU for 4 hours. Previous work from our lab and others has shown that these particular conditions of HU slow replication forks but also induce firing of new origins/forks (PMID: 25843623, 18079179). Based on the SMLM data in **new Fig. 4b** (middle), upon HU treatment there is increased BRCA2 recruitment to nascent DNA. PLA in these conditions similarly shows increased BRCA2/PCNA colocalization upon low-dose HU treatment (**new Fig. 4c**). As this reviewer has pointed out, indeed it has been shown that BRCA2 is recruited to HU-induced stressed forks (PMIDs: 21565612, 29038466, 32357836, 29038425) at high doses (> or = 4mM) of HU for 4+ hours. We would like to emphasize that at least a subset of BRCA2 recruitment to low-dose HU-induced forks are to new forks in this low-dose HU setting, and that this particular recruitment is transcription-dependent. These results are supported by the fact that BRCA2 near pSer2 RNAP2 is increased upon this low dose HU treatment by SMLM (**Fig. 4b, right**).

We have added statistics on this graph, thank you for catching this error, it was inadvertently omitted from the final figure version.

5. Fig. 4B, here the authors state: “depletion of RNaseH2A alone, was comparable to the co-depletion of RNaseH2A with BRCA2, suggesting that RNaseH2A and BRCA2 likely work in the same pathway to suppress HO-TRCs at TTS”.

Here the authors do not show the relevant statistic comparison, i.e., between siBRCA2 vs siRNH2A alone or siRNH2A vs siRNH2A+siBRCA2 so this is a far reached conclusion. In addition, we do not know that these are HO-TRCs and where they take place (see comment 2).

We thank the Reviewer for pointing out this error. We have attached an updated graph below including the appropriate statistical analysis. However, we have decided to not include this data in the manuscript to prioritize the functional interaction between BRCA2/RNase H2 through other experiments. We have expanded on this explanation below in major point #6.

Rebuttal Figure 6 (Corresponding to OLD Fig. 4b). PLA of pSer2-RNAP2 and PCNA with the indicated siRNAs in U2OS cells. *p* values calculated using Mann-Whitney rank-sum t-test of 200 nuclei per condition from 3 biological replicates. Error bars = mean, std.

6. In Fig. 4D and 4E, the authors show that RNaseH2A colocalizes less with elongating pol II in BRCA2 depleted cells and its localization is increased after HU treatment. The authors conclude that both BRCA2 and RNaseH2A cooperate to mitigate toxic HO-TRC. Again, a far-fetched conclusion since they do not show any evidence that BRCA2 and RNaseH2A are in proximity in any conditions. In Fig. 5 they show ATRi exacerbates HO-TRC independent of BRCA2 status so it would be a good condition to test PLA probing RNaseH2A and BRCA2.

Moreover, describing these figures authors state that "both BRCA2 and RNaseH2A in FTE cells were more cytotoxic compared to FTEs treated with either siControl, siBRCA2, or siRNaseH2A alone". So, in Fig. 4B depleting RNaseH2A leads to the same outcome as siBRCA2+ siRNaseH2A leading the authors to state they likely function in the same pathway, and now in Fig. 4F, they show the opposite, namely, synthetic lethality, i.e., different pathways, as previously reported; which one is the right one? How do authors explain these discrepancies?

We thank the reviewer for this helpful suggestion. We now provide information elucidating the interplay between BRCA2 and RNase H2 upon induction of TRCs. Shown below and included in **new Fig. 4h-i** shows FTEs treated with low-dose of HU, which we have shown causes TRC enriched with pSer2-RNAP2 (**new Supplementary Fig. 5k**) as well as ATRi as suggested by this Reviewer (**new Fig. 3f**). In both of these conditions, colocalization of BRCA2 and RNaseH2A are increased by PLA. Interestingly, we also see BRCA2 has increased interaction with RNase H1 upon low-dose HU treatment:

Rebuttal Figure 7 (Corresponding to new Fig. 4h-i).

(h): PLA of BRCA2 and RNaseH2A or BRCA2 and RNase H1 in FTEs +/- 0.2mM HU for 4h, including single antibody controls for each PLA. Data presented shows 200 nuclei from 3 biological replicates. *p* values calculated using unpaired two-tailed t-tests between

conditions. Left: number of foci per nucleus. Right: average number of foci per nucleus per biological replicates. Error bars = mean, std. Scale bar = 10µm in representative pictures.

(i): PLA of BRCA2 and RNaseH2A or BRCA2 in FTEs +/- ATRi. Data presented shows 200 nuclei from 3 biological replicates. *p* values calculated using unpaired two-tailed t-tests between conditions. Left: number of foci per nucleus. Right: average number of foci per nucleus per biological replicates. Error bars = mean, std.

Next, we appreciate the Reviewer for pointing out the discrepancies in our argument between the epistatic change in TRCs in siBRCA2, siRNASEH2A, and KD of both with our cell survival assay in the same conditions. While previous work shows BRCA2- and RNase H2 are synthetic lethal (PMID: 29973717), and indeed we see this in our FTE cell model system double-KD of BRCA2 and RNASEH2A compromises cell survival more than either KD alone. However, we agree this data adds an unnecessary layer of complexity given that double-KD of BRCA2 and siRNASEH2A is epistatic to siRNASEH2A alone, as pointed out by this Reviewer. Indeed, there are many other reported functions of these proteins: BRCA2 in replication fork protection (PMIDs: 21565612, 29038466, 32357836, 29038425), RNase H2 in ribonucleotide processing (PMID: 30380406, 26903602), and both involved in transcription elongation (PMID: 35618715, 29386125). Ongoing work in our lab is focused on characterizing the molecular function of this complex. We believe the above data showing BRCA2-RH2 cooperating together to respond to TRCs is more appropriate for this manuscript, and trying to understand the functional mechanism of BRCA2 and RNase H2 by epistasis analysis is beyond the scope of this study.

7. Interestingly, the authors show that CDC7i, which inhibits dormant origin firing, reduce the proximity of elongating Pol II and PCNA only in BRCA2 depleted cells (Fig. 5D). This is an important experiment to relate dormant origin firing observed in BRCA2 deficiency and the possible link with HO transcription-replication conflicts. Again, it should be performed in RNaseH1 (or RH2) treatment conditions to confirm the presence of RNA-DNA hybrids.

Thank you for this suggestion. We have now performed PLA probing for TRCs in siRNASEH2A – treated cells +/- CDC7i. The results below show that CDC7i rescues TRCs induced by loss of RNASEH2A (new Fig. 4G).

G

Rebuttal Figure 19 (Corresponding to new Fig. 4g). PLA of pSer2-RNAP2 and PCNA in siControl or siRNASEH2A FTEs +/- CDC7i. Data presented shows 200 nuclei from 3 biological replicates. *p* values calculated using unpaired two-tailed t-tests. Left: number of foci per nucleus. Right: average number of foci per nucleus per biological replicates. Error bars = mean, std.

8. The authors state: "CDC7i treatment decreased origin density" in both control and BRCA2-deficient FTEs: We showed that decreased origin density caused by CDC7i in BRCA2-deficient cells can occur without compensatory increase in fork speed (Figures 5B and 5C). This suggests that a major role for BRCA2 in protecting stalled forks is independent of changes to dormant origin firing".

This interpretation seems too simplistic or wrong. Given the different functions of BRCA2, and the replication stress induced by BRCA2 KD already, I do not see how the authors can infer any function of BRCA2 based on these results. Nevertheless, the result with CDC7i is important so they should confirm that there is no impact of CDC7i on origin firing at TSS in BRCA2 deficient cells as shown in fig. 1B for HU. This is to discard the possibility that there is a global effect of the inhibitor.

Thank you for this suggestion. Indeed, performing Ok-seq on CDC7i-treated cells would be ideal and this is something we plan to do in future studies. However, as an alternative to the concern that the CDC7i could be shutting down canonical fork usage, we'd like to point the reviewer to pSer5 RNAP2 – PCNA PLA in new Fig. 3e, where we showed that in both BRCA2-proficient and -deficient cells there is not a further decrease in pSer5 RNAP2-PCNA signal, which we use as a proxy for forks originating from TSSs.

Additionally, we have performed SMLM on BRCA2-proficient and -deficient FTEs treated with CDC7i, pulse-labeled with EdU. Using SMLM analysis, we are able to quantify the total nuclear EdU levels per S-phase nucleus as previously described (PMID: 35365626). Indeed, CDC7i does not decrease global EdU levels in either BRCA2-WT or -KD cells. This new data has now been added to **new Supplementary Fig. 6b**.

Rebuttal Figure 8 (Corresponding to new Supplementary Fig. 6b).

Representative SMLM image shows EdU (yellow) of pulse-labeled FTEs +/- shBRCA2 +/- CDC7i. Scatterplot quantification measuring total EdU levels per nucleus. *p* values calculated using unpaired two-tailed t-tests of at least two biological replicates: WT-BRCA2(-)CDC7i, N=91; WT-BRCA2(+), N=96; KD-BRCA2(-)CDC7i, N=83; KD-BRCA2(+), N=75. Error bars = mean, std.

9. The conclusion that R-loops persist at HO-TRCs due to the inability of BRCA2 to recruit RNaseH2 to these sites is also far-fetched (lane 4 before Discussion), see point 6.

We believe we have sufficiently addressed this point in major point #6, and with our new data in Fig. 4d-g, Fig. 5a-c, and Supplementary Fig. 7a-b.

Minor:

-The error bars in the BRCA2 KD cells in Fig. 2E are quite large so the results here are not convincing. Doing the same experiment with PCNA instead of EdU as in the PLA setting might be a way to reduce the variability.

We thank this author for this suggestion; however, we were hoping to use EdU-SMLM as a complementary approach to our RNAP2-PCNA data, rather than directly repeat the same experiment with another technique, especially when we have found strong agreement between our PLA and SMLM data. Because our SMLM analysis captures the pair-correlation globally between the two species of interest throughout the entire nucleus (in this case EdU and RNAP2), the spread pair correlation amplitude can be quite variable. However, the strength of this is that we are capturing an average of several interactions throughout the nucleus. We hope this reviewer can be convinced by the very significant P-value despite the large error bars.

- It is not clear to me why they choose to compare BRCA2-mutated to p53-mutated HGSOc cases from TCGA.

We chose to compare BRCA2-mutated HGSOcs to p53-mutated HGSOcs because nearly all HGSOcs are *TP53*-mutated (PMIDs: 29061967, 17117391, 27898521, 31772167). Therefore, we probed into the impact of mutBRCA2 specifically on genes with somatic mutations compared to patients with the same tumor suppressor driver mutation (mut*TP53*).

- Only a single BRCA2 patient (out of 2) shows EMT pathway enriched

Thank you for this comment. We realize this is correlative but we prefer to emphasize that the genes with somatic mutations in one BRCA2 patient was **only** enriched for EMT. We hope our proposal and hypotheses for the impact of TRCs in the final paragraph of the discussion emphasizes a dynamic, transcriptome-dependent model for TRC-induced genomic instability.

- Pages do not have numbers

We have added page numbers to the revised manuscript.

REVIEWERS' COMMENTS

Reviewer #1 (Remarks to the Author):

The authors have adequately addressed my previous questions and concerns. The additional experiments and analysis have significantly contributed and further strengthened the validity of their conclusions.

The only point I am still not convinced is to whether the authors can be 100% sure/confident that the observed TRCs with elongating RNAPII at the TTS of "high volume" genes are exclusively HO- oriented TRCs as stated as a strong conclusion in the title and the results of the manuscript. The authors provide much evidence that this is likely the case but I still do not think that the authors have shown this directly with their assays. Thus, I would recommend to change the title to "Dormant origin firing promotes transcription-replication conflicts at transcription termination sites in response to BRCA2 deficiency" and more carefully discuss the orientation of the conflicts in the manuscript.

Based on my original comment 3, the authors have also changed the statistical analysis of the PLA data, but it seems that this was not implemented for all PLA datasets, especially the new PLA combinations, e.g Fig. 3j,k,l. Please check throughout the manuscript and keep it consistent.

I congratulate the authors to a great study and an important contribution to the field.

Reviewer #2 (Remarks to the Author):

My concerns have been addressed in a proper manner in this revised version.

Reviewer #3 (Remarks to the Author):

The authors have addressed most of the points raised by the reviewers and by doing so they have substantially improved the strength of the manuscript supporting their conclusions. They have also removed some confusing data that did not add value and needed further confirmation. My only concern is still the point raised by reviewer 2 (point 9) or point 4 reviewer 3 (myself). This is about stating that at low doses of HU, BRCA2 is enriched at 'active forks'. Extensive electron microscopy and IF studies have shown that replication forks at that dose or less are stalled, many of which are also reversed, although not collapsed. So, even if the authors observe BRCA2 enriched at nascent DNA under those conditions, as others have seen too, it does not mean BRCA2 is enriched at new active forks. BRCA2 could be accumulating at slowing forks but the turnover of BRCA2 being reduced. This statement should be rephrased to avoid confusion. Point 8 about the major role of BRCA2 being independent of dormant origins is also not convincing and should be rephrased.

Here is our point-by-point response to the Reviewers' Comments from the resubmission (Author's comments/rebuttal in blue):

Reviewer #1 (Remarks to the Author)

The authors have adequately addressed my previous questions and concerns. The additional experiments and analysis have significantly contributed and further strengthened the validity of their conclusions.

We thank this reviewer for their helpful suggestions.

The only point I am still not convinced is to whether the authors can be 100% sure/confident that the observed TRCs with elongating RNAPII at the TTS of "high volume" genes are exclusively HO- oriented TRCs as stated as a strong conclusion in the title and the results of the manuscript. The authors provide much evidence that this is likely the case but I still do not think that the authors have shown this directly with their assays. Thus, I would recommend to change the title to "Dormant origin firing promotes transcription-replication conflicts at transcription termination sites in response to BRCA2 deficiency" and more carefully discuss the orientation of the conflicts in the manuscript.

To reflect the novelty of our study and distinguish our study from others in the TRC/replication stress field, we feel strongly about keeping the title as is. To lesson the definitiveness of our findings, we put air-quotes over "head-on", to signify the high probability that our evidence points to exclusively HO-oriented TRCs. We would like to re-highlight evidence we provide for head-on TRCs caused by dormant origin firing in BRCA2-KD.

- We have provided new evidence that the dormant origins in BRCA2-KD that we identified by Ok-seq occur specifically at TTSs of high-volume genes, or just downstream of these TTSs. Our relative distance analysis calculation shows BRCA2-KD leads to increased origins oriented at TTSs (Fig.1e, Fig.1g). When we look specifically at all intragenic or intergenic origins in BRCA2-WT or BRCA2-KD cells (Fig.S3a), we specifically observe that BRCA2-KD intergenic origins have a statistically significant increased orientation at high-volume TTSs compared to BRCA2-WT intergenic origins (Fig.S3b), or intragenic origins in either BRCA2-WT or -KD (Fig.S3c).

- Interactions between pSer2 RNAP2 and components of the replisome (PCNA in PLA assays, EdU in STORM assays) differ in BRCA2-KD compared to pSer5 RNAP2 (Fig.2). We suspect if there were increased codirectional collisions (as opposed to HO-TRCs) between forks emanating from TSSs in BRCA2-KD, then we would also observe increased pSer5 RNAP2 signal. We don't see increased pSer5 RNAP2 signal under BRCA2-KD conditions, suggesting that co-directional fork TRCs cannot explain the differences we are seeing in BRCA2-KD conditions.
- Our DRIP-qPCR data indicates there are increased R-loops at high-volume TTSs of genes with new origins (Fig.5c), while the signal is not increased at the TSS or introns of these genes (Fig.S7a). If altered transcription or replication dynamics in BRCA2-KD cells were contributing to codirectional collisions, we suspect we would again see a statistically significant change in the R-loop signal at these sites.

Additionally, similar papers studying TRCs that have employed replication-sequencing and transcription-sequencing techniques have argued for head-on or codirection orientation of TRCs in various genetic backgrounds based the orientation of replication relative to transcription (PMIDs: 33553603, 29466339, 36002001, 35100392). Therefore, we believe our manuscript is similarly, if not more, convincing with the current tools used to study TRCs in genome-wide, endogenous human models. We believe the air quotes used in our title represent the statistically significant data we have provided for HO-TRCs, while nothing in biology is 100% known.

Based on my original comment 3, the authors have also changed the statistical analysis of the PLA data, but it seems that this was not implemented for all PLA datasets, especially the new PLA combinations, e.g Fig. 3j,k,l. Please check throughout the manuscript and keep it consistent.

The updated statistical analysis for Fig.3j-l was already included in Fig.S6a, S6d-e. The Reviewer may have missed this from our resubmission files. Several of the PLA experiments showing statistical analysis for biological replicates was put in the supplementary info to conserve space in the main figures.

I congratulate the authors to a great study and an important contribution to the field.

We thank this reviewer for their suggestions and feedback that has strengthened our study.

Reviewer #2 (Remarks to the Author)

My concerns have been addressed in a proper manner in this revised version.

We thank the reviewer for their suggestions which have increased the strength of our manuscript.

Reviewer #3 (Remarks to the Author)

The authors have addressed most of the points raised by the reviewers and by doing so they have substantially improved the strength of the manuscript supporting their conclusions.

We thank the reviewer for their comments that have strengthened our study.

They have also removed some confusing data that did not add value and needed further confirmation.

My only concern is still the point raised by reviewer 2 (point 9) or point 4 reviewer 3 (myself). This is about stating that at low doses of HU, BRCA2 is enriched at 'active forks'. Extensive electron microscopy and IF studies have shown that replication forks at that dose or less are stalled, many of which are also reversed, although not collapsed. So, even if the authors observe BRCA2 enriched at nascent DNA under those conditions, as others have seen too, it does not mean BRCA2 is enriched at new active forks. BRCA2 could be accumulating at slowing forks but the turnover of BRCA2 being reduced. This statement should be rephrased to avoid confusion.

While we agree that BRCA2 recruitment to PCNA could represent a mix of BRCA2 recruitment to stalled and new forks, we do not suspect that BRCA2 recruitment to nascent DNA seen by STORM would represent recruitment stalled forks, as stalled forks would unlikely incorporate new EdU in a 10' EdU pulse.

Point 8 about the major role of BRCA2 being independent of dormant origins is also not convincing and should be rephrased.

We have removed the discussion of this point in our 2nd revised text.